

# Enhancing physically based and distributed hydrological model calibration through internal state variable constraints

Frédéric Talbot[1], Jean-Daniel Sylvain[2], Guillaume Drolet[2], Annie Poulin[1], Richard Arsenault[1]

[1] Hydrology, Climate and Climate Change Laboratory, École de technologie supérieure, Université du Québec, Montréal, H3C 1K3, Canada

[2] Direction de la recherche forestière, Ministère des Ressources naturelles et des Forêts, Québec, G1P 3W8, Canada

*Correspondence to*: Frédéric Talbot (frederic.talbot.2@ens.etsmtl.ca)

**Abstract.** This study investigates the effectiveness of various calibration approaches within the Water Balance Simulation Model (WaSiM) to enhance the representation of hydrological variables. We assess the impact of three distinct configurations: Baseline (BL), Physical Groundwater Model (GW), and Physical Groundwater with Recharge Calibration (GW-RC) on the representation of hydrological variables. The analysis demonstrates that while traditional calibration primarily enhances streamflow prediction, integrating recharge and groundwater dynamics significantly refines the model's ability to depict subsurface processes. The GW-RC configuration, with minimal emphasis on recharge in the objective function, shows a marked improvement in representing both the spatial and seasonal variability of groundwater recharge, suggesting that even small and targeted calibration adjustments can significantly enhance the accuracy and realism of model outputs. Although this approach may reduce the model's flexibility in mirroring observed streamflow, it enhances the precision with which other hydrological processes are represented, providing a more accurate reflection of watershed dynamics. Our findings underscore the importance of multi-variable calibration frameworks, which incorporate both streamflow and internal hydrological variables, in developing robust models capable of adapting to anticipated hydrological shifts due to climate change. This approach provides a more accurate reflection of watershed dynamics and offers valuable insights for calibration strategies in hydrological modelling, water resource management and climate adaptation strategies.

## 1 Introduction

Accurately representing watershed processes under climate change remains a central challenge in the evolving field of hydrology (Persaud et al., 2020). Recent advances in hydrological modeling have offered valuable insights into water resource management and climate adaptation strategies (Xu et al., 2005; Chen et al., 2011; Wang et al., 2023). However, the complexity of watershed dynamics, especially in snow dominated catchments, necessitates models that can accurately simulate both surface and subsurface hydrological processes (Chu and Shirmohammadi, 2004; Farjad et al., 2016).

The need for detailed, physically based hydrological modeling goes beyond immediate concerns of water management and climate impact assessments. Groundwater dynamics play a critical role in forest health (Maitre et al., 1999; Jacobs, 2003), as stable water availability, shaped by hydrological processes, underpins forest ecosystem resilience (Cunningham et al., 2011;



Orellana et al., 2012). By enhancing the accuracy of groundwater simulation and recharge calibration, we can improve our ability to forecast forest growth and resilience under changing climatic conditions (Ford et al., 2011; Grant et al., 2013). This linkage underscores the importance of detailed hydrological modeling and aligns with broader environmental, economic, and ecological management goals aimed at sustaining forest productivity in the face of environmental change. Such integrative

approaches are vital as they provide the groundwork for informed decision-making in forest management, ensuring that forests continue to thrive (Vose et al., 2011; Sun et al., 2023).

The Water balance Simulation Model (WaSiM) (Schulla, 2021) is a distributed and physically based hydrological model that stands out for its complexity, fine spatial resolution and comprehensive approach to modeling key hydrological processes.

This capability is particularly advantageous for yielding reliable results in intermediate variables analysis within hydrological studies. Several studies exemplify the application of WaSiM for examining internal hydrological variables across diverse geographic settings and scenarios. For example, Jasper et al. (2006) analyzed summer soil water pattern shifts due to climatic changes, demonstrating that WaSiM could effectively model the substantial alterations in hydrological responses to varying climate scenarios. Natkhin et al. (2012) used WaSiM to differentiate the impacts of climate change and

forest growth dynamics on groundwater recharge in Northeast Germany. Similarly, two separate studies (Rößler and Löffler, 2010; Rössler et al., 2012) analyzed soil moisture dynamics using WaSiM, discussing the modeling potentials and limitations in high mountain catchments and the broader impact of climate on soil moisture. Bormann and Elfert (2010) investigated how land use changes influence various runoff generation processes such as surface runoff, interflow, and baseflow. Furthermore, Förster et al. (2017, 2018) conducted detailed comparisons of internal state variables with actual

forest measurements, including meteorological variables and snow cover dynamics, highlighting the refined capabilities of WaSiM to model complex interactions like snow cover and canopy interception. These studies collectively demonstrate the model's utility in capturing a wide range of hydrological variables.

Recent advances in hydrological modeling have revealed critical challenges in accurately representing watershed dynamics, particularly when calibrating hydrological models based solely on streamflow data (Mei *et al.*, 2023; Schäfer *et al.*, 2023; de

Lima Ferreira and da Paz, 2024; Pool *et al.*, 2024). While streamflow is a key indicator for capturing temporal fluctuations in water systems, it offers limited insights into the internal hydrological processes (Rajib *et al.*, 2018). This reliance on streamflow can result in models that perform well in reproducing observed flows but misrepresent underlying processes—a phenomenon known as equifinality, where different parameter sets produce the same outputs but for the wrong reasons (Kirchner, 2006; Yassin *et al.*, 2017; Acero Triana *et al.*, 2019; Mei *et al.*, 2023). Therefore, focusing only on streamflow in

model calibration can hide important differences in how hydrological processes are represented.

In pursuit of better representing hydrological processes at the catchment scale, several studies have explored hydrologic scaling and parameter transferability (Samaniego *et al.*, 2010, 2017; Mizukami *et al.*, 2017; Imhoff *et al.*, 2020). Notably, Samaniego *et al.* (2010) introduced the multiscale parameter regionalization to tackle overparameterization and the non-transferability of parameters across different scales. Ficchì et al. (2019) also proposed a model structure that considers flow



accuracy and fluxes match on different modelling timesteps, adjusting the structure and parameters to ensure robust simulation across various time scales. Additionally, Peters-Lidard *et al.* (2017) advocated for adopting the fourth paradigm of data-intensive science in hydrology, which leverages emerging datasets to refine our understanding of hydrological models and processes. This paradigm posits that advancements in computational science—considered a new methodological branch alongside empiricism, theory, and computational simulation—can revolutionize science through the intensive use of

data, facilitating the discovery and testing of theories and models. This approach emphasizes the integration of comprehensive datasets and computational tools into conventional scientific workflows, thereby enhancing the capacity for scientific innovation and synthesis in hydrology.

Recent studies have advocated for a shift towards integrating additional hydrological variables and data sources, such as remote sensing products and in-situ measurements, into the calibration process (Dembélé *et al.*, 2020; Meyer Oliveira *et al.*,

2021; Liu *et al.*, 2022; Mei *et al.*, 2023; Schäfer *et al.*, 2023; de Lima Ferreira and da Paz, 2024; Pool *et al.*, 2024). Mei *et al.* (2023) found that including gridded soil moisture alongside gauged streamflow improved evapotranspiration simulations across 20 catchments in the Lake Michigan watershed. Schäfer *et al.* (2023) used WaSiM to simulate the water balance of a forested catchment in Germany, showing that including plant-available water and evapotranspiration data significantly enhanced model accuracy. De Lima Ferreira and da Paz (2024) similarly improved model performance by incorporating

actual evapotranspiration estimates into a hydrological model of a Brazilian semi-arid basin, highlighting the benefits of multi-variable calibration and the need to test distinct data sources.

Although many studies have successfully used variables such as soil moisture, evapotranspiration, and groundwater head in model calibration, there remains a gap in understanding how other variables, like groundwater recharge, can improve the representation of hydrological processes. Addressing this gap is important for both the theoretical advancement of

hydrological sciences and the practical applications of water resource management, flood risk assessment, and climate change mitigation (Pradhan and Indu, 2019). By adopting a calibration approach that integrates a more holistic view of watershed processes, models become more reflective of complex hydrological interactions and gain robustness in the face of non-stationary climate conditions (Wang *et al.*, 2023). This enhanced process representation and strengthens confidence in model projections, making them more reliable for future applications.

In this study, we implement three distinct model configurations of the WaSiM hydrological model, configuration BL (baseline model), configuration GW (physical groundwater model), and configuration GW-RC (physical groundwater and recharge calibration model)—to investigate how integrating additional hydrological variables and different calibration approaches influence the representation of hydrological processes over a set of 34 catchments in Nordic conditions. Through comparative analysis of these configurations, we aim to expose the nuances in model performance and hydrological variable

representation, contributing to the ongoing debate on the best practices for hydrological model calibration.



## 2 Methods

### 2.1 Study area

This study examines 34 catchments in Southern Quebec, Canada, each with distinct physiographic and hydrometeorological features. The catchments range in size from 525 to 6,840 km² (see Fig. 1). These specific catchments were selected for their inclusion in the Hydroclimatic Atlas of Southern Québec (MDDELCC, 2022) due to the availability of comprehensive streamflow data and their representation of the diverse hydrological conditions prevalent throughout Southern Quebec. Selected catchments are unaffected by the presence of dams and reservoirs, preserving the natural integrity of hydrological processes.

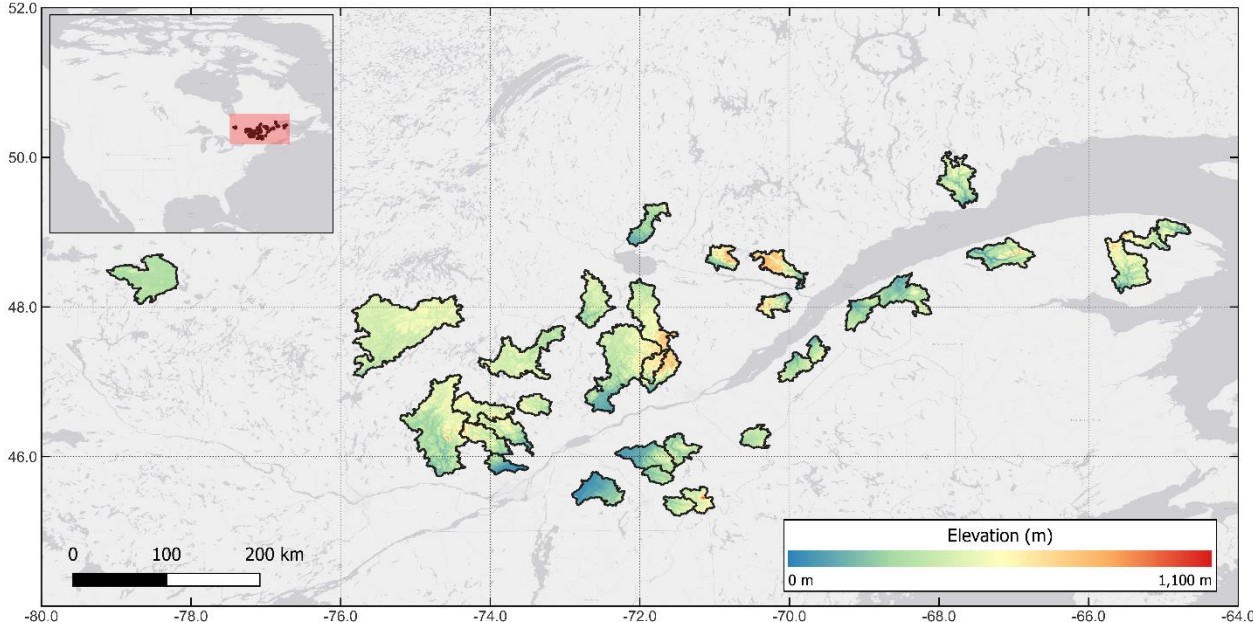

**Figure 1. Elevation map of study catchments in southern Quebec.**

The Köppen-Geiger Climate Classification designates most of the study area (28 catchments) as belonging to class Dfb (humid continental mild summer, wet all year), except a small part (six catchments) located in the northern portion that belongs to class Dfc (subarctic with cool summers and year-round precipitation) (Beck *et al.*, 2018). The region experiences four distinct seasons. Winters are characterized by frequent sub-freezing temperature and significant snowfall. As spring arrives, temperatures gradually rise, leading to significant snowmelt which, along with increasing rainfall, influences streamflow and water availability. Summer brings warmer temperatures, peaking in July, with rainfall remaining relatively high. Fall sees a gradual cooling and a transition from rain to increasing snowfall, setting the stage for another winter cycle. This climatic diversity induces complex hydrological processes at catchment scale, as the interplay between snowmelt and



precipitation patterns has a significant influence on streamflow and water availability. These dynamics are not unique to
Québec but are indicative of broader hydrological upheavals occurring across boreal regions globally under climate change.
To contextualize the environmental and hydrological setting of the selected catchments, Table 1 presents a synthesis of key
descriptors. The table shows the minimum and maximum values for a set of hydrological and geophysical characteristics for
each catchment, providing an at-a-glance perspective of the environmental variation within the study area.

**Table 1. Hydrological and geophysical characteristics of the study catchments.**

| Catchment characteristics | Minimum | Maximum |
|---|---|---|
| Area (km$^2$) | 525 | 6840 |
| Mean elevation (m) | 137 | 568 |
| Predominant soil type | Sandy loam | |
| Predominant land use | Coniferous forest and deciduous forest | |
| Annual total precipitation (mm) | 785 | 1547 |
| Annual extreme daily temperature (°C) | -37.7 | 28.6 |
| Annual streamflow (m$^3$ s$^{-1}$) | 10 | 130 |

## 2.2 Data

### 2.2.1 Hydrometeorological data

This study utilizes meteorological data, specifically total precipitation and mean temperature on a daily time step, sourced
from ECMWF's Reanalysis v5 (ERA5) (Hersbach *et al.*, 2020). These datasets effectively overcome the limitations of
observational data and have demonstrated performance on par with observational records in this region (Tarek *et al.*, 2020).
The collected meteorological data spans the period from 1981 to 2020.

Observed streamflow data from 1981 to 2010 was used, recorded at a daily resolution. This data was obtained from the
Hydroclimatic Atlas of Southern Québec (MDDELCC, 2022). The dataset contains occasional gaps, primarily during winter
months when ice cover and ice jams can significantly distort river flow measurements. To ensure the accuracy of the study,
these periods were excluded from the dataset.



### 2.2.2 Elevation data

A hydrologically conditioned digital surface model was derived from the NASA Shuttle Radar Topography Mission version 3.0 Global 1 (SRTM-DSM) to account for terrain elevation. The SRTM-DSM, originally boasting a spatial resolution of 30 meters at the equator, underwent resampling to 50 meters resolution and filtering using multiple moving average windows to mitigate the impact of local noise, which could lead to erroneous hydrological behaviours (MacMillan *et* al., 2000). To ensure hydrological consistency, we applied hydrological corrections based on data from provincial agencies (Géobase du réseau hydrographique du Québec (GRHQ) - Données Québec, 2016). The elevation values along established hydrological networks were adjusted downward by 5 meters burning the stream network into the digital surface model (DSM) with the SAGA GIS software (Conrad *et* al., 2015). The resulting DSM accurately captures the hydrological characteristics of the study area and is used for catchment delineation. Additionally, the DSM was resampled to spatial resolutions of 250 and 1000 meters. This resampling process was conducted to optimize computational efficiency while preserving the essential characteristics of the catchments. The minimum value resampling method was used to preserve hydrological connectivity within the study area.

Following this, the Tanalys software (Schulla, 2021) was used to generate key topographic layers, including slope, aspect, and river depth, all formatted for hydrological modeling within WaSiM.

### 2.2.3 Soil type data

To capture the spatial variability of soil hydraulic properties, we utilized the SIIGSOL 100 meters database (Sylvain et al., 2021), which provides information on soil composition. The SIIGSOL database provides detailed descriptions of the proportions of sand, clay, and silt within the soil profile (MRNF, 2022). In this study, we converted the reported proportions of sand, silt, and clay layers into soil texture classes based on the classification system of the United States Department of Agriculture (USDA). The USDA soil classification system categorizes soils into various texture classes such as loam, clay, sand, silt, and combinations thereof, which are determined based on the percentage composition of each type. This classification aids in understanding the soil's physical characteristics which are crucial factors in hydrological modeling and in predicting soil-water interactions in the studied catchments (Weil and Brady, 2017).

We derived soil hydraulic properties from generated soil type maps, using established relationships between soil texture classes and hydraulic parameters. For the soil type maps, WaSiM generates soil layers of specified thickness based on the control file settings. By default, if there is only one soil type present in the catchment, the soil depth is uniformly distributed throughout the entire area. To account for soil depth variability, we divided soil types into three distinct sections based on their relative elevation within catchment: narrow, normal, and deep. Pixels with elevations below the 33rd percentile were classified as deep, while those with elevations above the 66th percentile were classified as shallow. The remaining soil type rasters fell into the normal category. This classification was based on the imperfect but useful hypothesis that higher



elevations correspond to a closer proximity of bedrock to the surface, while lower elevations indicate a greater depth of soil cover in a post-glacial landscape (Akumu *et* al., 2016; Jeong *et* al., 2022).

### 2.2.4 Land use data

For land use attribution, we used the 2015 North American Land Change Monitoring System (NALCMS) 30 meters land cover dataset (Latifovic *et* al., 2012; Commission for Environmental Cooperation, 2020). The classification scheme used in this map adheres to the widely recognized Land Cover Classification System (LCCS) standard established by the Food and Agriculture Organization (FAO) of the United Nations. This standardized approach ensures the consistency and comparability of land cover information, enabling meaningful regional scale assessments and studies. The nearest neighbor resampling method was employed to align land use maps with the other raster maps used in WaSiM. Land use exerts a substantial influence on various hydrological parameters, and more specifically for the context of this study, it significantly affects parameters such as root distribution, vegetation cover fraction (VCF), roughness length (Z0), and albedo within the hydrological model. The distribution and characteristics of land cover types, ranging from forests to urban areas, directly impact these parameters, thereby influencing processes such as evapotranspiration, runoff, and infiltration.

### 2.2.5 Groundwater recharge data

In 2008, the Government of Quebec initiated the "Projets d'acquisition de connaissances sur les eaux souterraines" (PACES; roughly translated as "groundwater knowledge acquisition projects") (Carrier *et* al., 2013; Cloutier *et* al., 2013, 2015; Comeau *et* al., 2013; Larocque *et* al., 2013, 2015; Rouleau *et* al., 2013; Buffin-Bélanger *et* al., 2015; Lefebvre *et* al., 2015), aimed at enhancing understanding of the groundwater resources availability in Southern Quebec area. In addition to PACES, numerous studies conducted across the region have estimated groundwater recharge rates, which vary from 50 mm yr$^{-1}$ to over 500 mm yr$^{-1}$ depending on the location and years studied (Croteau *et* al., 2010; Chemingui *et* al., 2015; Larocque *et* al., 2019; Dubois *et* al., 2021; Boumaiza *et* al., 2022).

Of the 34 catchments in this study, fourteen were entirely or partially covered by the PACES project. Table 2 lists these catchments, detailing their areas, associated PACES region reports, the percentage of each catchment's area covered by PACES, and the mean and standard deviation of groundwater recharge for the areas covered.



**Table 2. PACES data coverage and groundwater recharge statistics for covered catchments.**

| Catchment name | Area (km²) | Region | Cover[1] | PACES recharge | |
|---|---|---|---|---|---|
| | | | | Mean (mm yr⁻¹) | Std. (mm yr⁻¹) |
| Matane | 1650 | Bas-Saint-Laurent | 31% | 179 | 78 |
| Rimouski | 1610 | Bas-Saint-Laurent | 29% | 213 | 81 |
| Des Trois-Pistoles | 932 | Bas-Saint-Laurent | 38% | 74 | 34 |
| Ouelle | 795 | Chaudière-Appalaches | 62% | 180 | 35 |
| Famine | 691 | Chaudière-Appalaches | 100% | 186 | 46 |
| Bécancour | 919 | Chaudière-Appalaches and Bécancour | 100% | 209 | 83 |
| Nicolet Sud-Ouest | 549 | Nicolet-Saint-François | 100% | 242 | 64 |
| Nicolet | 1540 | Nicolet-Saint-François | 95% | 224 | 82 |
| Noire | 1490 | Montérégie-Est | 93% | 133 | 98 |
| Rouge | 5460 | Outaouais | 26% | 310 | 40 |
| Kinojévis | 2590 | Abitibi-Témiscamingue | 55% | 172 | 87 |
| Petit Saguenay | 712 | Saguenay-Lac-Saint-Jean | 80% | 69 | 78 |
| Petite rivière Péribonca | 1090 | Saguenay-Lac-Saint-Jean | 29% | 142 | 103 |
| Valin | 746 | Saguenay-Lac-Saint-Jean | 73% | 221 | 85 |
| [1] Fraction of total catchment area covered by PACES data. | | | Median | 183 | 80 |

## 2.3 Hydrological modelling

### 2.3.1 WaSiM model

In this study, we employed WaSiM for hydrological modeling (Schulla, 2021). Hydrological processes were analyzed

through three specific configurations: BL (baseline), which serves as the standard comparison model; GW (physical groundwater model), which incorporates detailed groundwater dynamics; and GW-RC (physical groundwater model with constrained recharge), which further refines the groundwater variables by incorporating constrained recharge calibrations. Detailed descriptions of these configurations can be found in Sect. 2.4 of this study.

WaSiM consists of two versions: WaSiM version I, originally developed using the Topmodel approach for simulating

subsurface flows based on variable saturation areas, and WaSiM version II, an extended version with the process-oriented Richards approach. The Richards version, which considers hydraulic head gradients and detailed soil physical properties (pF-curve, k(u) function), was selected for this study due to its more physically based nature.

WaSiM follows a modular structure, composed of multiple sub-models that can be activated based on data availability and the specific research objectives. The model operates using a consistent time step, while internally employing flexible sub-




time steps to optimize computational efficiency. It accommodates both regular and irregular raster grids, enabling the analysis of diverse spatial configurations. During each time step, the sub-models are sequentially processed across the entire model grid, enabling parallelization to aid computational optimization and facilitate faster model execution.

One of the key process modules within WaSiM is the unsaturated zone model, which plays a crucial role in calculating various hydrological variables such as surface runoff, groundwater recharge, interflow, and baseflow. Interflow refers to
water moving laterally through the upper soil layers, contributing to streamflow, while baseflow is the portion of streamflow sustained by groundwater flow. These variables are essential for understanding the water balance and hydrological dynamics within the study area. Table 3 provides an overview of the hydrological model configuration used in this study.

**Table 3. Overview of WaSiM characteristics and sub-models used in this study.**

| Sub-model | Method | Reference |
|---|---|---|
| Meteorological interpolation | Inverse distance interpolation | (Shepard, 1968) |
| Potential evapotranspiration | Hamon approach | (Hamon, 1963) |
| Actual evapotranspiration | Richards equation using the Van Genuchten parameters | (Richards, 1931; van Genuchten, 1980) |
| Snow melt | Temperature-index approach | (Hock, 2003) |
| Interception | Classic bucket approach dependent on LAI | - |
| Lake modelling | Integrated approach to model natural and artificial lakes, considering interactions with unsaturated zone, routing, snow, evaporation, interception, and groundwater models. | - |
| Unsaturated zone flow | Richards equation using the Van Genuchten parameters | (Richards, 1931; van Genuchten, 1980) |
| Groundwater flow | Integrated two-dimensional groundwater model | - |
| Routing | Kinematic wave approach | (Lighthill and Whitham, 1955) |

Meteorological data interpolation was an essential step in the hydrological modeling process. The chosen hydrological model, WaSiM, performed the interpolation of daily precipitation and temperature inputs between ERA5 points. For each simulation, the model creates grids that incorporate the interpolated meteorological values at the model's spatial resolution, effectively representing the climatic conditions for each individual pixel. The inverse distance weighting method was used as recommended by WaSiM model description report (Schulla, 2021).

**2.3.2 Calibration parameters**

Calibration of WaSiM involved the optimization of 17 parameters, selected in accordance with WaSiM documentation (Schulla, 2021), while the remaining parameters in the control file were set to their default values. Table 4 provides a





detailed description of upper and lower limits set for calibrating the 17 parameters in WaSiM, with each parameter adjusted
to two decimal places within the specified calibration range.

**Table 4. Description of the parameters used for the calibration of WaSiM.**

| No. | Code | Description | Unit | Sub-Model | Range |
|---|---|---|---|---|---|
| 1 | $k_D$ | Storage coefficient for surface runoff | h | Unsaturated zone | [1, 25] |
| 2 | $k_H$ | Storage coefficient for interflow | h | Unsaturated zone | [1, 25] |
| 3 | $d_r$ | Drainage density for interflow | $m^{-1}$ | Unsaturated zone | [1, 50] |
| 4 | $QD_{Snow}$ | Fraction of surface runoff on snow melt | - | Unsaturated zone | [0.1, 1] |
| 5 | $c_0$ | Degree-Day factor | $mm °C^{-1} d^{-1}$ | Snow | [0, 3] |
| 6 | $T_0$ | Temperature limit for snow melt | °C | Snow | [-4, 4] |
| 7 | $T_{R/S}$ | Transition temperature snow/rain | °C | Snow | [-4, 4] |
| 8 | $C_{WH}$ | Water storage capacity of snow | - | Snow | [0.1, 0.3] |
| 9 | $C_{rfr}$ | Coefficient for refreezing | - | Snow | [0.1, 1] |
| 10 | $f_{i,summer}$ | Summer correction factors for ETP | - | Evapotranspiration | [0.1, 2] |
| 11 | $f_{i,fall}$ | Fall correction factors for ETP | - | Evapotranspiration | [0.1, 2] |
| 12 | $f_{i,winter}$ | Winter correction factors for ETP | - | Evapotranspiration | [0.1, 2] |
| 13 | $f_{i,spring}$ | Spring correction factors for ETP | - | Evapotranspiration | [0.1, 2] |
| 14 | $K_{rec}$ | Recession constant for hydraulic conductivity | - | Soil table | [0.1, 0.99] |
| 15 | $d_z$[a] | Soil layer thickness | - | Soil table | [0.8, 1.4] |
| 16A | $K_B$ | Storage coefficient for base flow | m | Unsaturated zone | [0.1, 8] |
| 17A | $Q_0$ | Scaling factor for base flow | $mm\ h^{-1}$ | Unsaturated zone | [0.1, 5] |
| 16B | Kol[b] | Colmation of the river links | - | Input grid | [1, 100] |
| 17B | $K_{XY}$[c] | Saturated horizontal conductivity (x-y-direction) | $m\ s^{-1}$ | Input grid | [0.2, 4] |

[a] Calibration coefficient, ranging from 0.8 to 1.4, is applied to adjust the total soil depth, which is predetermined to be 8 meters for shallow, 14 meters for normal, and 20 meters for deep soil conditions.

[b] Calibration coefficient, ranging from 0.8 to 1.4, is applied to adjust the colmation grid, which is predetermined to be $1x10^{-6}$.

[c] A calibration coefficient, ranging from 0.2 to 4, is applied to adjust the saturated horizontal conductivity grid, which is predetermined to be $4x10^{-5}$ $m\ s^{-1}$.

Parameters 16A ($K_B$) and 17A ($Q_0$) are calibrated in the configuration BL when groundwater model is not activated and instead uses a conceptual approach to compute groundwater flow within the unsaturated zone sub-model. Groundwater flow is assessed using Eq. (1) (Schulla, 2021), which calculates baseflow as a function of several parameters including the scaling factor for baseflow ($Q_0$) and the recession constant for baseflow ($K_B$).


$$Q_B = Q_0 * K_s * e^{(h_{GW} - h_{geo,0})/K_B}, \tag{1}$$



where $Q_B$ is baseflow (m s$^{-1}$), $Q_0$ is a scaling factor for baseflow, $K_s$ is the saturated hydraulic conductivity (m s$^{-1}$), $h_{GW}$ is the groundwater table height (m), $h_{geo,0}$ is the geodetic altitude of the soil surface (m) and $K_B$ is the recession constant for baseflow (m).

In the configurations used in GW and GW-RC, which activate groundwater model, parameters $16_A$ and $17_A$ are replaced by parameters $16_B$ and $17_B$ to obtain a more physically based representation of groundwater processes. Parameters $16_B$ and $17_B$ adjust values associated to two input grids that allow to account for the colmation of the river links and saturated horizontal conductivity. This distinction ensures a consistent number of calibrated parameters across all configurations, facilitating an unbiased comparison of model performance.

### 2.3.3 Model optimization

Parameters optimization was performed independently for each catchment through the dynamically dimensioned search algorithm (DDS; (Tolson and Shoemaker, 2007)), following the recommendation of Arsenault *et* al. (2014). This algorithm is specifically designed for efficiently calibrating complex hydrological models with a large parameter range given a finite computing budget. During optimization, it dynamically adapts its search strategy based on the number of evaluations performed and performance metrics. To manage computational demands effectively while ensuring thorough exploration of the parameter space, a two-phase calibration strategy was employed, albeit the approaches differ for the constrained groundwater configurations.

Initially, 1000 simulations were performed for each catchment at a broader spatial resolution (1000 meters) using a broader range of values for each parameter (Table 4). This phase aimed to identify an approximation of the optimal values for each parameter. Subsequently, these values were used to initialize the second calibration step at a finer spatial resolution (250 meters). This sequential calibration strategy allows to refine the model's performance progressively. By first identifying a set of parameters that achieves reasonable model performance at a coarser scale, we then fine-tune the model at a higher resolution to enhance the spatial distribution of hydrological simulations.

The objective functions used vary by configuration: For BL and GW, the objective is to optimize the Kling-Gupta Efficiency (KGE, (Kling *et* al., 2012)), as discussed in Sect. 2.5.1. Conversely, the GW-RC configuration employs a modified objective function that seeks to optimize KGE and constrain groundwater recharge rates and variability. This approach is described in Sect. 2.4.3 and Sect. 2.5.2.

The study employed split-sample test (SST) framework for the parameter optimization assessment. This widely used approach involves dividing the available data into two sets: one for calibrating the model and the other for validating its performance on unseen time periods. The calibration period (2000-2009) and the validation period (1990-1999) were chosen based on the availability of comprehensive and reliable hydrological data. A five year spin-up period was performed before each simulation to allow the model to reach a stable state, eliminating the influence of unstable initial conditions on the model's performance metrics. However, data gaps were noted for three catchments: Croche, Petit Saguenay, and Sainte-Marguerite Nord-Est. Specifically, Croche lacked data from 2001 to 2004, Petit Saguenay from 2000 to 2010, and Sainte-



Marguerite Nord-Est from 1998 to 2010. To accommodate these gaps, adjustments were made to the calibration and
       validation periods for the affected catchments. The calibration periods were shortened to later years: 1995 to 1999 for
       Croche and Petit Saguenay, and 1992 to 1996 for Sainte-Marguerite Nord-Est. Correspondingly, the validation periods were
       adjusted to precede the missing data: 1991 to 1994 for Croche, 1986 to 1994 for Petit Saguenay, and 1986 to 1991 for
       Sainte-Marguerite Nord-Est.

**2.4 Model configurations**

       The primary objective of this research is to examine how different model configurations influence the representation of
       hydrological processes. To ensure a consistent comparison of model configuration and calibration, we designed a modelling
       framework that allow to compare three configurations that incrementally incorporate more complex hydrological variables.

       **2.4.1 Baseline**

The first configuration (BL), serving as baseline configuration, employs the standard calibration of the model without
       activating the groundwater module. This configuration is aligned with the traditional application of WaSiM, where the focus
       is predominantly on streamflow, and groundwater flow is modeled using Eq. (1) within the unsaturated zone sub-model. This
       configuration is comparable to what has been frequently adopted in numerous studies, providing a common basis for
       comparative analysis (Rössler *et al.*, 2012; Förster *et al.*, 2018; Markhali *et al.*, 2022; Valencia Giraldo *et al.*, 2023).

**2.4.2 Physical groundwater module**

       The second configuration, GW (physical groundwater), marks a departure from the BL configuration by activating WaSiM's
       groundwater module. This adjustment allows for groundwater flow to be simulated within a designated sub-model,
       transitioning from a conceptual to a more physically based representation. This configuration, used in numerous studies
       (Bormann and Elfert, 2010; Natkhin *et al.*, 2012; Gädeke *et al.*, 2014; Schäfer *et al.*, 2023), is recommended by the WaSiM
documentation for catchments where groundwater dynamics play a pivotal role in the hydrological cycle, particularly in
       lowland areas with extensive sediment layers.

       **2.4.3 Physical groundwater module and constrained recharge**

       For configuration GW-RC (physical groundwater and constrained recharge), we incorporate groundwater recharge into the
       calibration process to achieve a better representation of hydrological variables such as baseflow, interflow, and runoff. By
introducing recharge into the calibration, we restrict hyperplane exploration and ensure that the model's representation of the
       hydrological cycle is more accurately simulating groundwater recharge dynamics. This is particularly useful if model
       hydrological variables are an important input to another analysis or process, such as for better understanding groundwater
       movement and evolution under climate change for certain types of vegetation, for example.



The calibration for configuration GW-RC was conducted in two distinct phases. The initial phase involved defining new
parameter ranges for parameters that impact baseflow (dr, QDSnow, Krec, Kol, Kxy). We therefore first conducted 200
evaluations at a spatial resolution of 1000 meters, followed by 50 evaluations at 250 meters using the objective function
presented in Eq. (6). Essentially, the aim here is to constrain the parameter set to a single value that performs well overall
and provides realistic internal variables. Similar approaches have been used in studies such as Duethmann *et* al. (2024),
which underscores the benefits of integrating Landsat-derived land surface temperature (Ts) data into model calibration.
Landsat, a series of Earth-observing satellites, provides crucial Ts data used in this study. By including satellite-derived Ts,
the study demonstrated improvements in the model's ability to capture spatial anomalies and ecosystem stress responses,
while maintaining streamflow accuracy, illustrating the advantages of multi-variable constraints in model calibration.

Following pre-calibration at both spatial resolutions, the resulting calibrated parameter sets were analyzed to define new
parameter ranges for the calibration phase. This analysis involved adjusting the minimum and maximum values of
parameters influencing baseflow ($d_r$, $QD_{Snow}$, $K_{rec}$, Kol, Kxy) by ±10% to establish new calibration ranges.

In the second and most important calibration phase, the process continued with the adjusted parameter ranges, employing a
less restrictive objective function (Eq. (7)) to better accommodate uncertainties in the recharge data. This phase involved a
comprehensive series of 1000 evaluations at 1000 meters and 50 at 250 meters resolutions. The modified objective function
primarily emphasized the KGE while incorporating the standard deviation of recharge at a reduced influence of 4%. This
modification was crucial to allow the model flexibility to adapt the groundwater recharge rate according to the specific
hydrological characteristics and precipitation patterns of each catchment. Given that the initial recharge rate of 250 mm yr$^{-1}$
was a preliminary estimate and not necessarily reflective of individual catchment conditions, this approach enabled a more
tailored calibration.

Table 5 shows an overview of the three methods to ease comparisons between configurations.



**Table 5. Summary of configurations**

| Settings | BL | GW | GW–RC |
|---|---|---|---|
| Groundwater Modelling | Conceptual within unsaturated zone sub-model | Physically based within the groundwater sub-model | Physically based within the groundwater sub-model |
| Calibration Parameters | 17 parameters (including KB and Q0) | 17 parameters (including Kol and Kxy) | 17 parameters (including Kol and Kxy) |
| Precalibration | N/A | N/A | 1. 200 simulations at 1000 meters 2. 50 simulations at 250 meters |
| Calibration | 1. 1000 simulations at 1000 meters 2. 50 simulations at 250 meters | 1. 1000 simulations at 1000 meters 2. 50 simulations at 250 meters | 1. 1000 simulations at 1000 meters 2. 50 simulations at 250 meters |
| Objective Function | Kling-Gupta efficiency | Kling-Gupta efficiency | Constrained Kling-Gupta efficiency |

## 2.5 Performance assessments

### 2.5.1 Kling-Gupta efficiency

The KGE (Kling *et* al., 2012) was chosen as the objective function to assess the model's performance during the calibration
process.

The KGE is computed using Eq. (2):

$$KGE = 1 - \sqrt{(r-1)^2 + (\beta-1)^2 + (\gamma-1)^2}, \quad (2)$$

where *r* is the correlation coefficient, calculated as:

$$r = \frac{\sum_{i=1}^{n}(O_i-\bar{O})*(S_i-\bar{S})}{\sqrt{\sum_{i=1}^{n}(O_i-\bar{O})^2 * \sum_{i=1}^{n}(S_i-\bar{S})^2}}, \quad (3)$$

here, $O_i$ and $S_i$ are the daily observed and daily simulated streamflow values, respectively, for each day $i$ in the series. $\bar{O}$ and
$\bar{S}$ are the average values of these daily observed and simulated streamflow across the entire series.

$\beta$ is the bias ratio, defined as:

$$\beta = \frac{\mu_{sim}}{\mu_{obs}}, \quad (4)$$

A bias ratio of 1 indicates no bias. Values less than 1 suggest underestimation by the model, while values greater than 1
suggest overestimation.

$\gamma$ is the variability ratio, calculated as:





$$\gamma = \frac{\sigma_{sim}/\mu_{sim}}{\sigma_{obs}/\mu_{obs}}. \tag{5}$$

The variability ratio assesses how effectively the model reproduces the variability in streamflow. It considers differences in the amplitude of variations in simulated and observed streamflow. Again, values less than 1 suggest underestimation by the model, while values greater than 1 suggest overestimation.

The resulting KGE values range from $-\infty$ to 1, where a KGE of 1 indicates a perfect match between observed and simulated streamflow. According to Knoben *et* al. (2019), a KGE value greater than -0.41 indicates that the model's performance is an improvement over using the mean flow as a benchmark.

### 2.5.2 Constrained Kling-Gupta efficiency

An arbitrary baseline groundwater recharge rate of 250 mm yr$^{-1}$ and a standard deviation of 80 mm yr$^{-1}$ have been established as representative benchmarks for the studied catchments. These values are based on PACES data and additional studies conducted in Quebec, as described in Sect. 2.2.5. The objective function for the pre-calibration of configuration GW-RC, outlined in Eq. (6), aims to balance KGE with these established recharge metrics. Specifically, the function assigns a weight of 70% to KGE, 20% to the annual recharge standard deviation, and 10% to the mean annual recharge. This specific weighting was chosen based on preliminary tests, where various weight combinations were evaluated on a test catchment. This objective function was designed to ensure both the quantity and variability of recharge were realistically modeled without sacrificing performance in terms of overall streamflow quality through the KGE.

The objective function employed in the pre-calibration of GW-RC configuration is formulated as follows:

$$Precalibration\ function = 1 - (0.7 * KGE + 0.2 * \lceil\sigma_{r_{sim}} - 0.08\rceil + 0.1 * \lceil\overline{r_{sim}} - 0.25\rceil), \tag{6}$$

where $\sigma_{r_{sim}}$ is the simulated annual recharge standard deviation (m yr$^{-1}$), $\overline{r_{sim}}$ is the simulated mean annual recharge (m yr$^{-1}$) and $KGE$ is the Kling-Gupta efficiency.

Groundwater recharge simulations were performed at the pixel level, ensuring detailed local representation. The simulated mean annual recharge reflects the average amount of recharge occurring annually across the entire catchment during the calibration period. Similarly, the simulated annual standard deviation quantifies the variability in annual recharge across all pixels within the catchment during the same period. Introducing pixel level standard deviation helps in curbing extreme values in groundwater recharge, thus stabilizing the simulation outputs. The mean annual recharge is employed to verify that the model accurately captures the overall recharge volume expected for the study area.

For the main calibration phase of the GW-RC configuration, the objective function is simplified to focus more intensively on streamflow accuracy:

$$Calibration\ function = 1 - (0.96 * KGE + 0.04 * \lceil\sigma_{r_{sim}} - 0.08\rceil), \tag{7}$$

where $\sigma_{r_{sim}}$ is the annual recharge standard deviation (m yr$^{-1}$) and $KGE$ is the Kling-Gupta efficiency.





## 2.6 Statistical analysis

To assess the performance of the hydrological model configurations, statistical analyses were conducted to compare calibration and validation performance across different configurations. The primary metric used was the KGE, which

evaluates the accuracy of simulated streamflow against observed data. The performance metrics were analyzed for each configuration during both the calibration period (2000-2009) and validation period (1990-1999), ensuring robust evaluation across varying hydrological conditions.

All statistical comparisons were made using the Kruskal-Wallis test, a non-parametric method chosen due to its suitability for non-normally distributed data. This test was employed to detect significant differences in the performance and

hydrological responses between the model configurations. Where significant differences were identified, multiple comparison post-hoc tests were conducted to ascertain the specific pairs of configurations that differed significantly.

Pearson's correlation coefficients were used to explore the influence of calibration parameters on hydrological variables. This statistical approach provided insights into how variations in parameter settings across different configurations could affect the representation of hydrological processes like surface runoff, interflow, and groundwater recharge.

## 380 3 Results

### 3.1 Calibration and validation performance

Throughout the calibration (2000-2009) and validation (1990-1999) periods, all configurations yielded KGE values above 0.5. Calibration and validation performances were very similar, with a deviation less than 5%, demonstrating the robustness of the simulations. KGE values for all catchments and configurations, for both the calibration and validation periods, are

presented in Table A1.

Figure 2 reveals a clear trend where catchments with high KGE values during calibration tend to maintain similar performance during validation. This consistency underpins the robustness of the configurations across different validation periods. During the validation period, median KGE values were higher for configurations BL (0.824) and GW (0.830) compared to GW-RC (0.770), demonstrating superior performance in the models without groundwater recharge constraints.

However, GW-RC demonstrates more consistent KGE values between calibration and validation, suggesting it may offer more stability in model performance despite its slightly lower KGE scores.





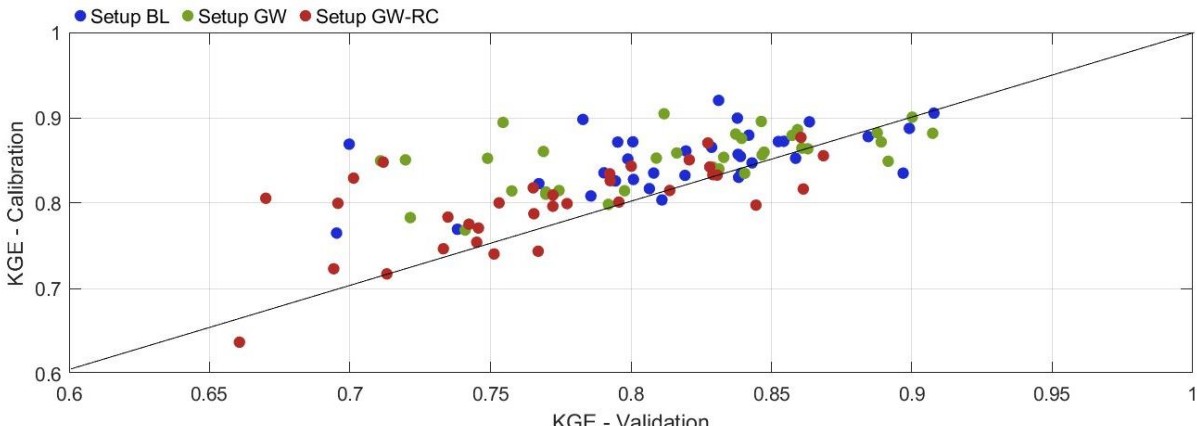

**Figure 2. Comparison of Kling-Gupta Efficiency values between calibration and validation periods for three configurations. Each point represents a catchment, color-coded by configuration: Configuration BL (blue), Configuration GW (green), and**

**Configuration GW-RC (red). The line represents a one-to-one relationship where calibration and validation KGE values are equal. Points below the line indicate better performance in the validation phase compared to calibration, while those above the line show a decline in performance from calibration to validation.**

It is important to note that the KGE values for configuration GW-RC are slightly lower than those from configurations BL and GW, which is expected given the supplementary constraints imposed during calibration.

**3.2 Hydrological variables analysis**

This section delves into the simulated hydrological variables, examining their range and distribution across the various model configurations during the calibration and validation periods. The variables in focus include surface runoff, baseflow, interflow, groundwater recharge, and actual evapotranspiration (ETa).

Figure 3 illustrates the annual totals (means for groundwater level and soil moisture) for simulated hydrological variables for

both calibration and validation periods and for all catchments. Notably, there is a consistency in the distribution of hydrological variables of each model configuration between the calibration and validation periods, which allows us to focus our detailed analysis solely on the validation period for conciseness.

A comparative assessment reveals distinct patterns in the simulated hydrological variables among the configurations. Specifically, configuration GW-RC simulates higher surface runoff and lower interflow, and infiltration compared to

configurations BL and GW. Conversely, configuration BL is characterized by higher actual evapotranspiration, lower groundwater recharge, and a higher groundwater level. Configuration GW shares similarities with both configuration BL (in terms of runoff, interflow, and infiltration) and configuration GW-RC (regarding baseflow, groundwater recharge, actual evapotranspiration, and groundwater level).





**Figure 3. Boxplots illustrating annual totals (means for groundwater level and soil moisture) variability of model internal variables. These boxplots detail the variability of key hydrological variables modeled with the different configurations, for calibration and validation periods and for all catchments.**

Figure 4 presents the proportional distribution of surface runoff, baseflow, interflow, and actual evapotranspiration for the three hydrological model configurations (BL, GW, and GW-RC). The charts effectively compare the relative contribution of each process to the total water cycle within the modeled catchments.

The figure highlights that configuration GW-RC simulates a notably higher proportion of surface runoff (21%) and baseflow (17%) with a lower proportion of interflow (20%). Conversely, configuration BL has a higher proportion of actual evapotranspiration (47%) and less baseflow (11%). Finally, configuration GW has similarities with both BL (surface runoff and interflow) and GW-RC (baseflow and actual evapotranspiration) configurations.





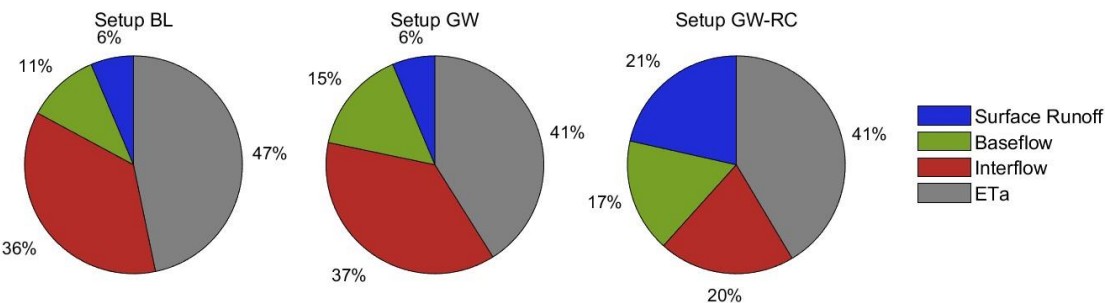


**Figure 4. Proportional distributions of key hydrological variables for the BL, GW and GW-RC hydrological model configurations.**

Table 6 shows that the observed similarities in surface runoff and interflow between configurations BL and GW are substantiated by statistical significance in their mean groupings. Furthermore, the parallels drawn between configurations GW and GW-RC in terms of actual evapotranspiration and groundwater recharge are also supported by significant statistical

evidence. However, the apparent similarity in baseflow between configurations GW and GW-RC does not hold statistical significance.

**Table 6. Statistical analysis of the differences in estimated hydrological variables from the three configurations BL, GW and GW-RC.**

| Hydrological Variables | BL vs. GW | BL vs. GW-RC | GW vs. GW-RC |
|---|---|---|---|
| Surface runoff | 0 | 1 | 1 |
| Baseflow | 1 | 1 | 1 |
| Interflow | 0 | 1 | 1 |
| Actual evapotranspiration | 1 | 1 | 0 |
| Groundwater recharge | 1 | 1 | 0 |

(Not Different = 0; Different = 1)

Figure 5 illustrates the distribution of key hydrological variables for the 34 catchments and for each configuration. Consistent trends in hydrological responses are observed across the catchments for each model configuration. For instance, configuration GW-RC typically shows higher runoff and lower interflow values across most catchments. Similarly, configuration BL consistently reports higher actual evapotranspiration and lower groundwater recharge. These patterns, initially observed in Fig. 3 and Fig. 4, are corroborated across most catchments, aligning with the statistical findings

presented in Table 6.





**Figure 5. Boxplots of annual values for key hydrological variables predicted by WaSiM for the 34 catchments and three configurations.**





Figure 6 presents the relationships between key hydrological variables and selected calibration parameters. All subplots
show high levels of correlations, shedding light on how varying the magnitude of calibration parameters influence model
behavior. Notably, surface runoff exhibits a strong correlation (r = 0.899) with the parameter QDsnow, which determines the
proportion of runoff from snowmelt. Actual evapotranspiration shows a notable correlation (r = 0.683) with the correction
factors for potential evapotranspiration (ETp), and interflow is similarly strongly linked (r = 0.801) to the drainage density
parameter. Baseflow and groundwater recharge display a strong correlation (r = 0.850) across all configurations. For
configurations GW and GW-RC, baseflow is inversely but strongly correlated (r = -0.875) with drainage density, whereas in
configuration BL, it correlates (r = 0.715) with the scaling factor for baseflow, Q0. It is also observed that configuration
GW-RC generally has a higher QDsnow parameter and a lower drainage density. Additionally, configuration BL is
characterized by larger correction factors for ETp.

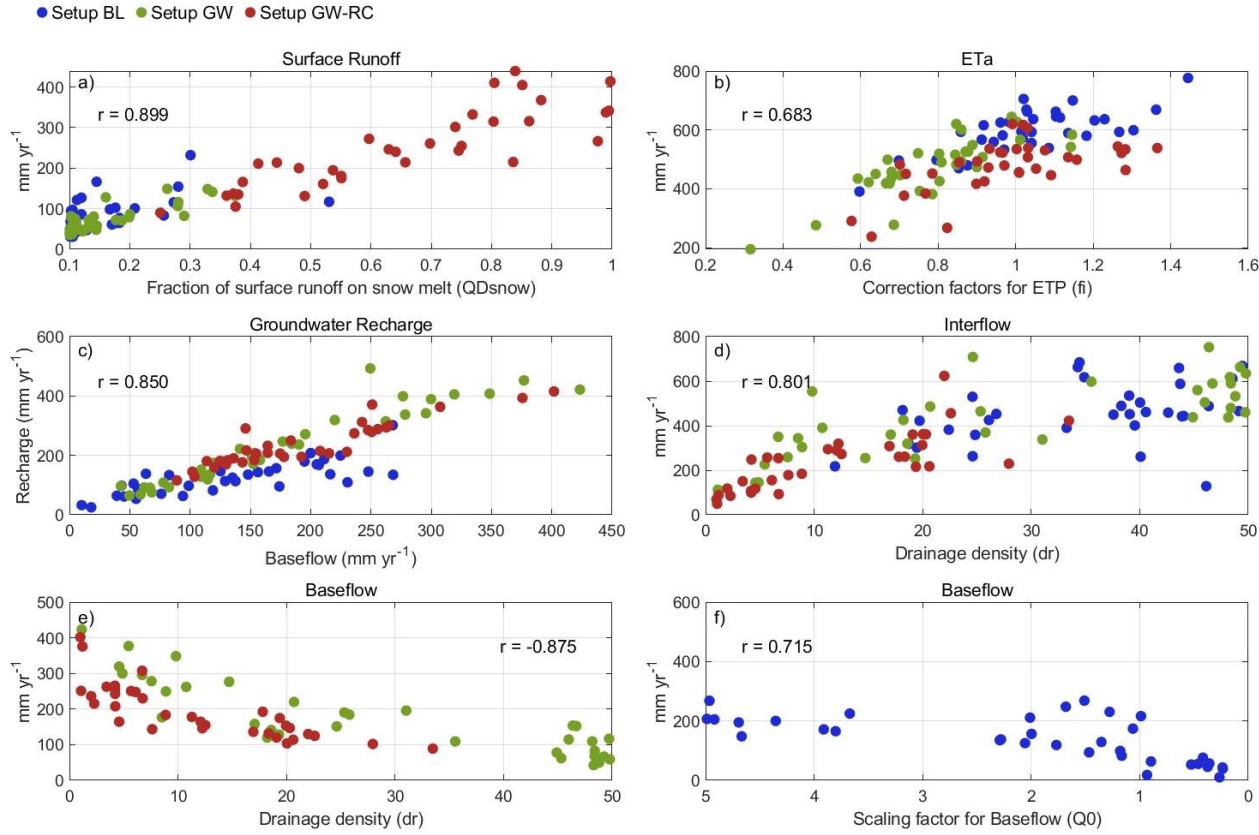

**Figure 6. Correlations between key hydrological variables and calibration parameters for three model configurations.**



## 3.3 In-depth analysis of the Matane catchment

This section explores the temporal dynamics of streamflow and hydrological variables in the Matane catchment, which was selected as a representative example from the study's catchments. Figure 7 contrasts observed and simulated streamflow for the Matane catchment during both calibration and validation periods, across the three configurations This figure highlights
the high similarity in the simulated streamflow between all configurations for both calibration and validation periods with the largest differences happening between April and July. This period aligns with seasonal high flows due to snowmelt. While configurations BL and GW exhibit higher KGE values during these periods, configuration GW-RC demonstrates a slightly reduced performance, in alignment with observations from Sect. 3.1. Nonetheless, all configurations show good performance, highlighting their robustness throughout both the calibration and validation periods.

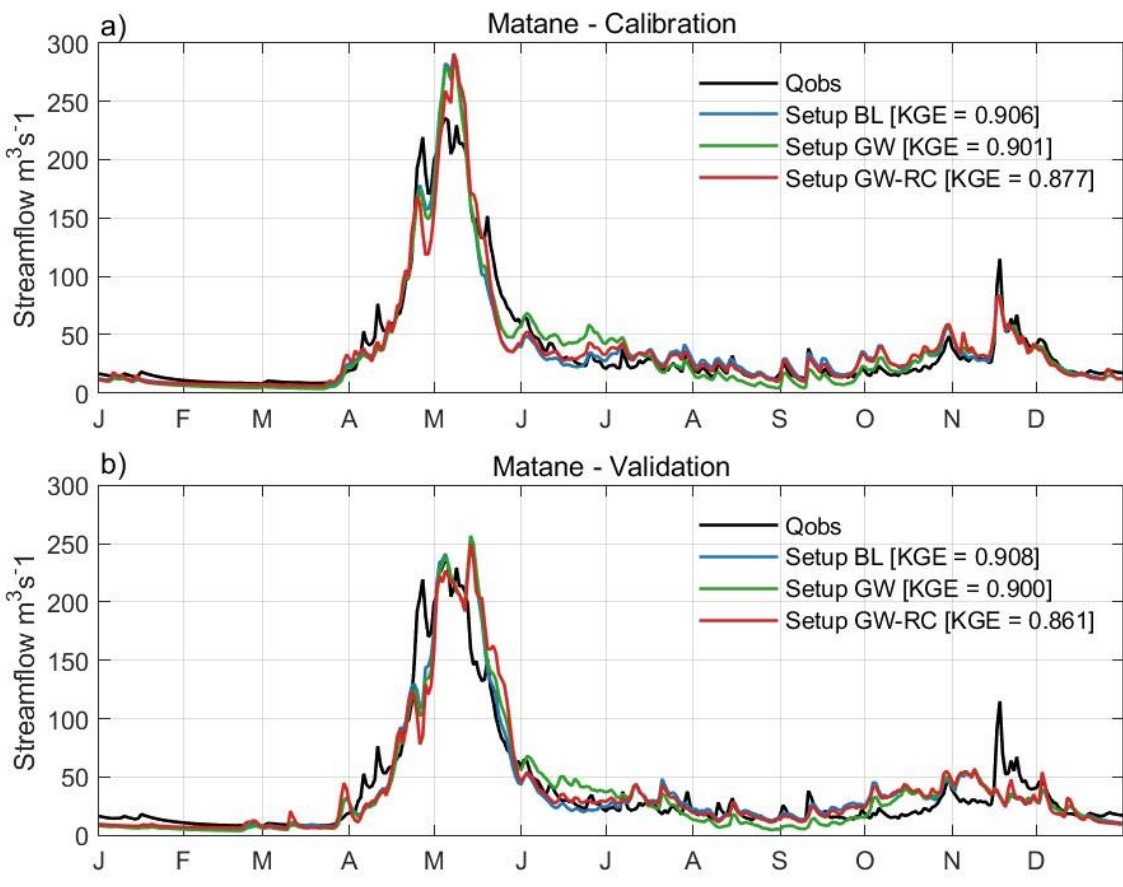


**Figure 7. Comparative hydrographs for Matane catchment showing modeling results from the three configurations as well as streamflow observations (Qobs).**





Figure 8 reveals consistent patterns in hydrological variable behavior across all configurations during both the calibration and validation periods. Consequently, the following discussions will focus primarily on the validation period. Generally,

interflow is the major contributor to simulated streamflow in configurations BL and GW throughout the year. In contrast, configuration GW-RC is characterized by a significant increase in surface runoff during the seasonal high flow and high precipitation periods in the fall, while predominantly exhibiting interflow contributions during other times of the year. Configuration GW-RC is also marked by higher levels of surface runoff and baseflow, but lower interflow compared to the other configurations. Configuration BL is distinguished by having the highest levels of annual actual evapotranspiration.

Configuration GW aligns closely with configuration BL in terms of interflow, surface runoff, and baseflow, demonstrating similar hydrological dynamics between these two configurations.

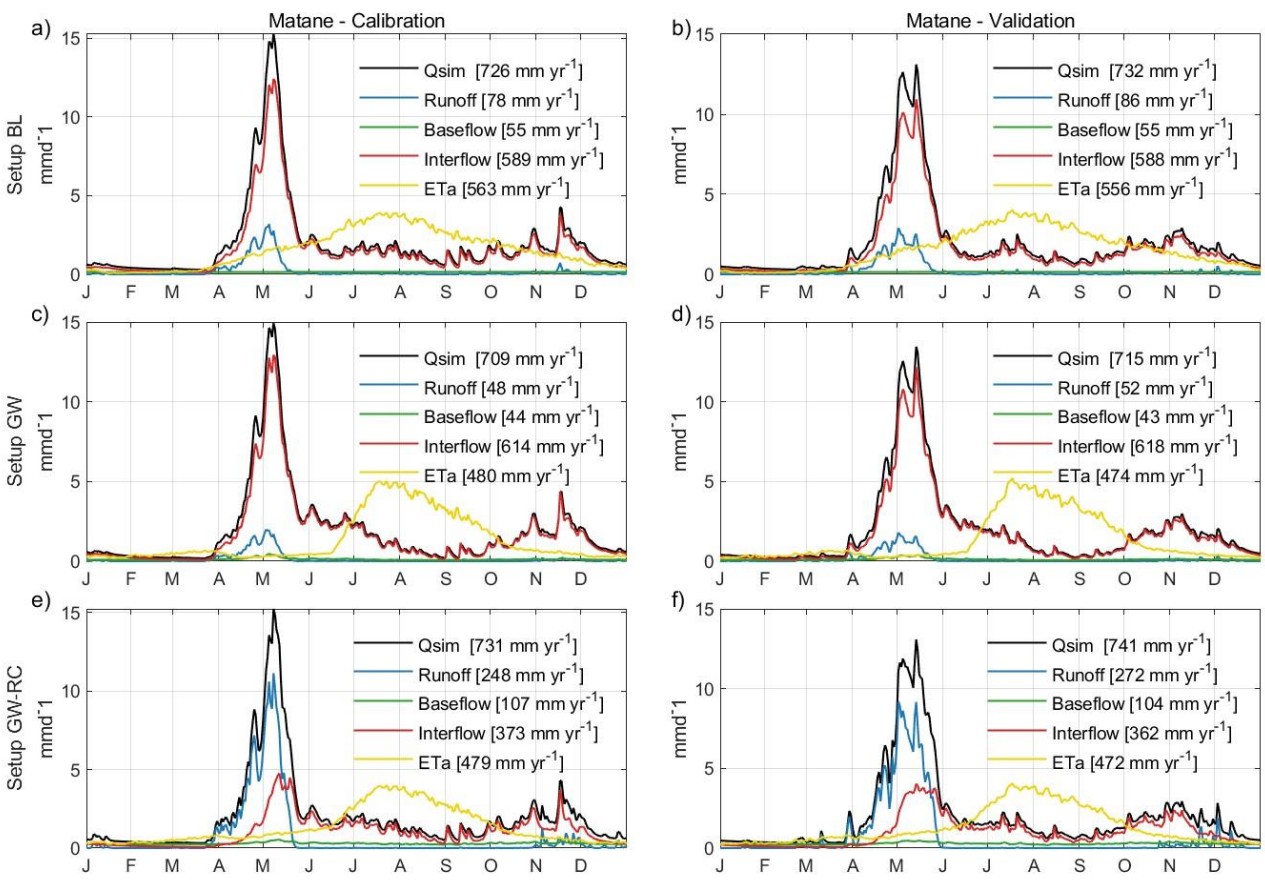

**Figure 8. Detailed hydrological variable hydrograph for Matane catchment during both the calibration and validation phases and for the three configurations. Calibration results are shown in panels (a), (c), and (e) for Configurations BL, GW, and GW-RC,**

**respectively, while validation results are depicted in panels (b), (d), and (f). These hydrographs demonstrate how baseflow,**



**interflow and runoff contribute to total streamflow throughout the year, with noted annual totals provided for a comprehensive comparison.**

Figure 9 reveals seasonal variations that correlate with hydrological responses to climatic conditions. Surface runoff and interflow differ significantly during periods of high flow, typically driven by snowmelt. Configurations BL and GW
primarily attribute high flows to interflow, whereas configuration GW-RC reflects these peaks with increased surface runoff. Groundwater recharge in configuration BL exhibits more pronounced seasonal fluctuations compared to the patterns observed in configurations GW and GW-RC. Similarly, configuration BL maintains a consistent baseflow year-round, unlike configurations GW and GW-RC, which show seasonal baseflow variations. In terms of actual evapotranspiration, configuration BL consistently exhibits higher rates in the spring and fall, GW peaks during the summer, and GW-RC
displays a pattern that blends characteristics of both BL and GW across different seasons.

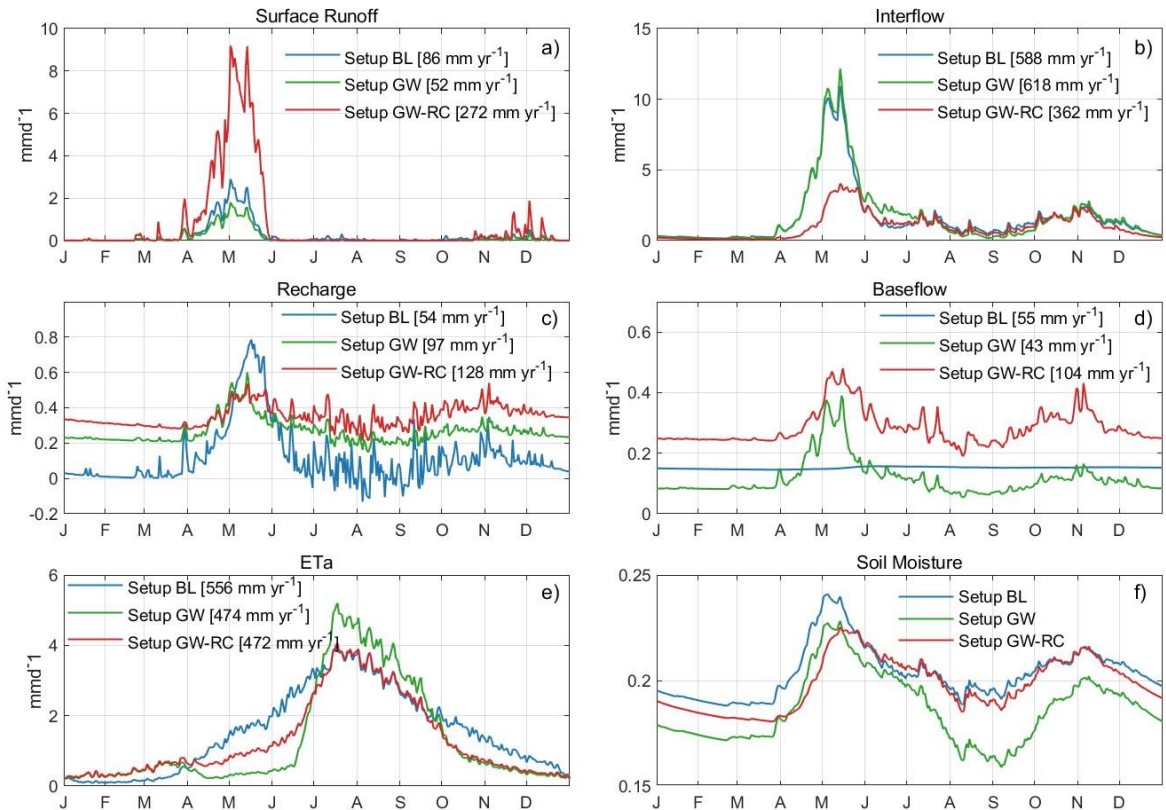

**Figure 9. Seasonal distribution of hydrological variables in the Matane catchment. This figure visualizes the annual distribution of key hydrological variables across the three configurations throughout the year.**



### 3.4 Groundwater recharge analysis

This section evaluates groundwater recharge, focusing on the influence of differing model configurations within WaSiM. Figure 9 panel C illustrates the daily groundwater recharge in the Matane catchment for each configuration. A common seasonal pattern is evident across all configurations: recharge decreases in winter, rises significantly during snowmelt, and then exhibits marked variability throughout summer and autumn. Notably, configuration GW-RC shows a lower dynamic range during snowmelt compared to configurations BL and GW, which exhibit more pronounced peaks. Throughout the

winter, summer, and autumn months, configuration GW-RC consistently shows higher recharge rates than the other configurations. The trends observed in the Matane catchment are also representative of the behaviors seen across all studied catchments

Further analysis involves distributed maps of annual recharge (Fig. C1), calculated at the pixel level for seven catchments, comparing PACES data with model outputs. Visually, configurations GW and GW-RC show recharge distributions that are

more consistent with the PACES dataset, suggesting a better spatial accuracy in these configurations compared to BL.

Figure 10 presents the boxplots of the annual recharge of each pixel for all configurations and the PACES data for the seven catchments. Configuration GW-RC's recharge estimates generally align more closely with the PACES data, indicating its ability in capturing the annual recharge dynamics at a finer spatial resolution. The other configurations follow, with GW also showing a reasonable approximation of PACES data, whereas BL appears less representative.

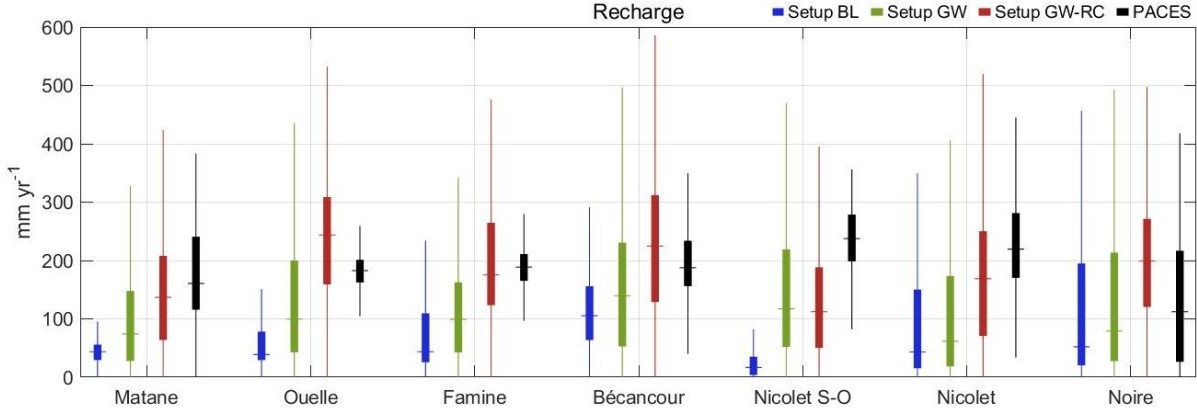


**Figure 10. Distributions of annual groundwater recharge across seven catchments, for the three configurations and the PACES data. Each boxplot represents the range and distribution of pixel-level annual recharge values in millimeters per year (mm yr⁻¹), with the central line indicating the median and whiskers extending to the 25th and 75th percentiles.**





## 4 Discussion

### 4.1 Performance against representation

This study aimed to analyze how varying model configurations affect the representation of hydrological variables estimated by WaSiM. Through the comparative analysis of three distinct calibration configurations, BL (baseline model), GW (activated groundwater simulation), and GW-RC (groundwater simulation and recharge calibration), this study provides insights into how internal hydrological processes are represented in a physically based model.

KGE values were consistently higher for the BL and GW configurations compared to GW-RC during both calibration and validation periods. Configuration GW-RC's modestly lower performance on KGE is reflective of its calibration not solely focusing on optimizing KGE but also in incorporating a broader suite of hydrological dynamics.

This finding aligns with prior research, which suggests that adding constraints to model parameters can often improve the representation of other hydrological processes, such as groundwater dynamics and soil moisture, albeit at the cost of lower

validation performance. For instance, Yassin *et* al. (2017) emphasized that incorporating additional data, such as from the Gravity Recovery and Climate Experiment (GRACE), can lead to more comprehensive and physically realistic model. Similarly, Dembélé *et* al. (2020) showed that incorporating spatial patterns from satellite data significantly improve the model's representation of soil moisture and evapotranspiration. Similarly, Bouaziz *et* al. (2021) found substantial disparities in internal process representation among models calibrated to the same streamflow data, highlighting the limitations of

relying solely on discharge data for model validation. Lastly, Pool *et* al. (2024) demonstrated that incorporating variables such as actual evapotranspiration and total water storage alongside discharge in model calibration can significantly enhance the simulation accuracy for these variables.

### 4.2 Hydrological variables analysis

Regarding the distribution of hydrological variables, configuration BL demonstrated the highest actual evapotranspiration

rates, alongside the lowest groundwater recharge and baseflow. Conversely, GW-RC was noted for the highest surface runoff and the lowest interflow. Configuration GW demonstrated characteristics that were intermediate between the other two configurations, mirroring BL in terms of interflow and surface runoff while aligning more closely with GW-RC in terms of groundwater recharge, actual evapotranspiration, and baseflow.

Baseflow is closely correlated (r = -0.875) with the drainage density parameter (scaling parameter for interflow) for

configurations GW and GW-RC. The constrained parameter range in configuration GW-RC explains the minor differences in baseflow rates observed between these configurations. In contrast, the baseflow in configuration BL is significantly correlated (r = 0.715) with the scaling factor for baseflow. The differences in groundwater recharge and baseflow across the configurations can be primarily attributed to the activation of the groundwater flow sub-model. In WaSiM, the simulation of groundwater processes can either follow a more conceptual or physically based pathway. Our results indicated that GW and





GW-RC, which incorporate more complex mechanisms between groundwater and surface processes, lead to more dynamic and possibly more accurate representations of baseflow and recharge dynamics.

The disparities in interflow of configuration GW-RC are mostly linked to the restricted calibration of the drainage density parameter with a strong correlation (r = 0.801) noted between interflow rates and the parameter value, highlighting how constraining the groundwater recharge during calibration can impact other hydrological variables like interflow. Similarly,

variations in surface runoff in configuration GW-RC are tied to the calibration restrictions on the 'QDsnow' parameter (fraction of surface runoff on snow melt), which is strongly correlated (r = 0.899) with surface runoff rates, indicating a significant control over this hydrological variable. Also, configuration GW-RC showed the highest value for 'QDsnow' parameter and the lowest value for the drainage density parameter consequently leading to the highest surface runoff and lowest interflow rates. This observation indicates that interflow is a flexible variable within the model, with configurations

BL and GW appearing to prioritize it over surface runoff and baseflow. This prioritization allows the optimization algorithm greater latitude to enhance performance metrics like KGE and more accurately reproduce observed streamflow patterns. Conversely, configuration GW-RC, constrained by groundwater recharge, tends to prioritize baseflow and surface runoff. While this approach may reduce the model's flexibility in mirroring observed streamflow, it enhances the precision with which other hydrological processes are represented as detailed in Sect. 4.3. The same trend was found for the Matane

catchment, underlining the broader applicability of these findings across different geographical contexts. Such a representation offers essential information that can be pivotal for water management strategies.

**4.3 Pinpointing the optimal model configuration**

The differences in surface runoff during the snowmelt season across configurations can be largely attributed to the parameter QDsnow. WaSiM employs a singular parameter (QDsnow) to account for surface runoff from snowmelt. This parameter is

calibrated between 0 and 1, and its precise setting critically influences the model's surface runoff predictions.

Analysis of Fig. 8 reveals that configurations BL and GW exhibit lower surface runoff from snowmelt, where melted snow predominantly percolates into the soil, contributing to interflow rather than surface runoff. This behavior is unexpected because, in fully frozen soil conditions, significant surface runoff is typically anticipated due to reduced infiltration.

Conversely, configuration GW-RC, which integrates groundwater recharge into the calibration process, follows a more

typical hydrological pattern. Higher surface runoff is observed at the onset of snowmelt, gradually decreasing as infiltration and interflow increase when the soil thaws. This progression aligns with the expected hydrological responses in frozen terrains, illustrating how the inclusion of groundwater recharge can improve the model's simulation of seasonal transitions. This trend of higher surface runoff during snowmelt was observed consistently across all catchments in the study, with detailed figures provided in the supplementary material (Fig. S1 to Fig. S32). Configuration GW-RC showed increased

surface runoff during the snowmelt period compared to the other configurations. However, for 11 out of the 34 catchments, the surface runoff results were notably elevated. Figure B1 illustrates an example where nearly all of the spring discharge was attributed to surface runoff, suggesting that the value assigned to the QDsnow parameter, when set too close to 1, may



lead to an overestimation of runoff. Careful calibration of this parameter is essential to avoid misrepresentations in the hydrological processes.

The analysis of groundwater recharge, as detailed in Sect. 3.4, reveals significant differences in seasonal dynamics and spatial distribution among the configurations. Notably, GW-RC displays less dynamic recharge rates during the snowmelt period compared to configurations BL and GW. This is indicative of a distinct interplay between surface runoff and infiltration processes within configuration GW-RC, where higher surface runoff during the spring results in reduced infiltration. Additionally, GW-RC exhibits higher recharge rates during summer, fall, and winter, with a peak in fall.

Spatial analysis through distributed maps and boxplot representations of annual recharge (Fig. S1 and Fig. 10) demonstrates that configuration GW-RC's recharge estimates align more closely with PACES data than the other configurations. Similarities between both datasets suggests that configuration GW-RC provides a more precise representation of spatial variability in recharge, indicating its enhanced ability to capture the real-world spatial distribution of recharge processes across diverse landscapes effectively.

Supporting these observations, Chemingui *et* al. (2015) found the average recharge rates across different seasons at three locations in the "des Anglais" catchment. The numbers retrieve in their work closely align with those simulated by the GW-RC configuration: winter (58 vs 50 mm), spring (58 vs 54 mm), summer (92 vs 60 mm), and fall (52 vs 72 mm).

Furthermore, Rivard *et* al. (2014) utilized the HELP infiltration model to simulate recharge for a catchment in Eastern Canada, reporting average recharge rates of 67 mm in winter, 62 mm in spring, 27 mm in summer, and 76 mm in fall. These
findings align with our results from configuration GW-RC, which also show peak recharge occurring in fall rather than in spring, differentiating it from the other configurations. Configuration GW aligns less precisely with these specific seasonal patterns, with a peak recharge in spring, but still outperforms BL in terms of matching the documented recharge rates from PACES.

Recharge rates from GW-RC align well with the PACES spatial distribution and compare favorably with observed seasonal
fluctuations in the literature. Overall, GW-RC's alignment with empirical data and its ability to simulate hydrological processes more accurately make it a preferable model configuration for studying and predicting hydrological dynamics under varied climatic conditions.

In this study, the GW-RC configuration demonstrated that assigning a minor weight to recharge in the objective function can significantly enhance WaSiM's capability to represent hydrological variables accurately, even with non-exact prior recharge
data. This approach underscores, again, the potential of leveraging prior information to refine model outputs, suggesting that even a modest emphasis on recharge within the calibration framework can lead to substantial improvements in model realism. This finding is particularly noteworthy as it implies that effective model calibration does not necessarily require precise initial recharge estimates if the calibration process is appropriately managed. It also points to the broader applicability of using informed yet flexible calibration strategies to improve hydrological models under varied conditions,
highlighting a path forward for enhancing model accuracy with limited prior data.





## 4.4 Practical implications, general applicability and limitations

The practical implications of this research extend beyond hydrological process modeling. Integrating groundwater recharge into model calibration, as demonstrated in the GW-RC configuration, offers a more comprehensive approach to representing key hydrological variables. This approach is particularly valuable for improving predictions of water resources under varying
climate conditions, as it enhances the accuracy of inputs critical to models of forest growth (Ford *et* al., 2011; Grant *et* al., 2013). As climate change continues to alter hydrological dynamics, the reliance on physically based models becomes crucial. These models are favored over conceptual ones or even machine learning based models because they can be adapted more readily to varying conditions, ensuring more robust predictions under climate change scenarios. For example, a strong recent trend is the use of deep learning architectures in hydrological modelling (Kratzert et al. 2018, 2019; Arsenault et al. 2023).
These models simulate streamflow with generally better accuracy than traditional hydrological models, but they lack any mechanism to investigate internal and intermediate hydrological variables. Such adaptability is also critical for effective water resource management and mitigation of climate impacts (Wilby, 2005; Ludwig *et* al., 2009; Poulin *et* al., 2011).

This research emphasizes the need to calibrate hydrological models using not only streamflow but also other variables such as groundwater recharge. This approach aligns with findings from other studies such as Yassin *et* al. (2017) and Dembélé *et*
al. (2020), which advocate for multi-objective calibrations that enhance model reliability across different hydrological variables. By integrating measurements from diverse sources such as satellite data and in-situ measurements, models can avoid the pitfalls of calibration based solely on streamflow, which might not capture the full spectrum of watershed dynamics. Bouaziz *et* al. (2021) further illustrate this point by demonstrating how hydrological models calibrated solely on streamflow can yield differing results when validated against other hydrological variables, underscoring the risk of
equifinality where different parameter sets produce similar results for streamflow but diverge for other variables. Without proper constraints—such as incorporating groundwater recharge into the calibration process—models may produce seemingly accurate streamflow simulations while failing to capture the underlying hydrological processes like configurations BL and GW.

The methodology developed in this study has broad applicability beyond the specific context of Southern Québec. This
approach can be valuable in a variety of geographic regions and hydrological settings, given similar contexts of equifinality (i.e. more processes and parameters than the data can support). Moreover, this multi-variable calibration method can enhance the accuracy of other distributed hydrological models by improving the representation of groundwater recharge related processes. Similar calibration techniques using remote-sensing data have been applied successfully in different settings, demonstrating that incorporating additional hydrological variables in calibration improves model performance.
Nevertheless, it is crucial to address the limitations of this study. The models' performance in replicating hydrological processes like soil frost impacts and its implications on runoff and recharge remain unknown. Future studies would benefit from incorporating field measurements alongside a broader range of climatic and hydrological conditions. Expanding the



research to include different geographic regions with similar soil and climate characteristics could significantly enhance the validation and applicability of the findings.

Moreover, the uncertainty inherent in modeling, especially with configurations that involve complex interactions of multiple variables, poses a continuous challenge. The study's reliance on specific data sets like PACES also introduces potential biases that could influence the generalizability of the findings. It's essential for future research to explore these limitations, perhaps by expanding the range of observational data used for model validation.

In terms of practical implementations and further research, continuing to refine the calibration of hydrological models to
include diverse hydrological variables can enhance their utility in real-world applications. Such efforts will help in developing more accurate flood forecasting models, improving water resource management strategies, and crafting more effective climate adaptation measures for forest, agricultural and anthropogenic ecosystems.

Overall, while this study lays a solid foundation for using advanced calibration techniques in hydrological modeling, the journey towards fully reliable and universally applicable hydrological models continues.

**5 Conclusion**

This study examined the nuances of hydrological modeling under different calibration settings using WaSiM model across 34 catchments classified under climate zones Dfb and Dfc in Eastern North America. By implementing three distinct model configurations, BL (baseline model), GW (physical groundwater model), and GW-RC (physical groundwater and recharge calibration model), this research has demonstrated that incorporating groundwater recharge alongside streamflow during
calibration process leads to a more accurate representation of hydrological variables.

The results indicate that the GW-RC configuration, enhanced with groundwater recharge calibration, aligns more closely with estimated groundwater recharge rates, thereby providing a more precise representation of groundwater behaviour both spatially and seasonally. The study also underscores the importance of extending calibration beyond traditional streamflow metrics to include other hydrological variables like groundwater recharge. This approach helps to mitigate the risks of
equifinality.

Given the successful application of these methodologies within Eastern North American catchments, it presents an intriguing premise for their applicability to other geographical areas with similar hydrological contexts. Further research could explore how these calibration techniques perform under different hydrological conditions, potentially broadening our understanding of these relationships.





## Appendix A

Table A1. Kling-Gupta efficiency values across studied catchments during calibration and validation periods, for the three calibrations configurations. Each row corresponds to a specific catchment, identified by its basin number.

| Catchment | | Calibration | | | Validation | | |
|---|---|---|---|---|---|---|---|
| Code | Name | BL | GW | GW-RC | BL | GW | GW-RC |
| 10802 | Bonaventure | 0.835 | 0.849 | 0.817 | 0.897 | 0.892 | 0.861 |
| 20404 | York | 0.847 | 0.872 | 0.815 | 0.843 | 0.889 | 0.814 |
| 20602 | Dartmouth | 0.888 | 0.882 | 0.842 | 0.899 | 0.907 | 0.828 |
| 21601 | Matane | 0.906 | 0.901 | 0.877 | 0.908 | 0.900 | 0.861 |
| 22003 | Rimouski | 0.920 | 0.905 | 0.870 | 0.831 | 0.812 | 0.827 |
| 22301 | Des Trois-Pistoles | 0.898 | 0.895 | 0.848 | 0.783 | 0.754 | 0.712 |
| 22507 | Du Loup | 0.872 | 0.852 | 0.800 | 0.795 | 0.749 | 0.696 |
| 22704 | Ouelle | 0.900 | 0.896 | 0.834 | 0.838 | 0.846 | 0.792 |
| 23422 | Famine | 0.826 | 0.814 | 0.754 | 0.794 | 0.798 | 0.745 |
| 24003 | Bécancour | 0.861 | 0.859 | 0.788 | 0.820 | 0.816 | 0.765 |
| 30101 | Nicolet Sud-Ouest | 0.828 | 0.810 | 0.771 | 0.801 | 0.770 | 0.746 |
| 30103 | Nicolet | 0.804 | 0.799 | 0.744 | 0.811 | 0.792 | 0.767 |
| 30234 | Eaton | 0.769 | 0.768 | 0.637 | 0.738 | 0.741 | 0.661 |
| 30282 | Au Saumon | 0.836 | 0.815 | 0.717 | 0.790 | 0.774 | 0.713 |
| 30304 | Noire | 0.823 | 0.813 | 0.723 | 0.767 | 0.770 | 0.694 |
| 40204 | Rouge | 0.830 | 0.842 | 0.798 | 0.838 | 0.829 | 0.844 |
| 40830 | Gatineau | 0.817 | 0.840 | 0.796 | 0.807 | 0.831 | 0.772 |
| 43012 | Kinojévis | 0.765 | 0.850 | 0.784 | 0.695 | 0.711 | 0.735 |
| 50119 | Mattawin | 0.852 | 0.814 | 0.740 | 0.799 | 0.758 | 0.751 |
| 50135 | Croche | 0.835 | 0.835 | 0.833 | 0.839 | 0.840 | 0.831 |
| 50144 | Vermillon | 0.835 | 0.853 | 0.747 | 0.808 | 0.809 | 0.733 |
| 50304 | Batiscan | 0.878 | 0.856 | 0.801 | 0.884 | 0.847 | 0.796 |
| 50408 | Sainte-Anne | 0.872 | 0.860 | 0.833 | 0.852 | 0.847 | 0.829 |
| 50409 | Bras du Nord | 0.853 | 0.864 | 0.856 | 0.859 | 0.863 | 0.869 |
| 52212 | Ouareau | 0.855 | 0.881 | 0.818 | 0.839 | 0.837 | 0.765 |
| 52219 | L'Assomption | 0.865 | 0.886 | 0.851 | 0.829 | 0.859 | 0.821 |
| 52233 | De l'Achigan | 0.869 | 0.851 | 0.829 | 0.700 | 0.720 | 0.701 |
| 52805 | Du Loup | 0.808 | 0.783 | 0.800 | 0.786 | 0.721 | 0.753 |
| 60101 | Petit Saguenay | 0.895 | 0.879 | 0.843 | 0.864 | 0.857 | 0.800 |
| 61801 | Petite rivière Péribonca | 0.833 | 0.876 | 0.775 | 0.819 | 0.839 | 0.742 |
| 61502 | Métabetchouane | 0.872 | 0.861 | 0.806 | 0.801 | 0.769 | 0.670 |




| 62701 | Valin | 0.880 | 0.882 | 0.826 | 0.842 | 0.888 | 0.793 |
| 62802 | Sainte-Marguerite Nord-Est | 0.872 | 0.854 | 0.810 | 0.854 | 0.833 | 0.772 |
| 71401 | Godbout | 0.857 | 0.864 | 0.799 | 0.838 | 0.861 | 0.777 |

**Appendix B**

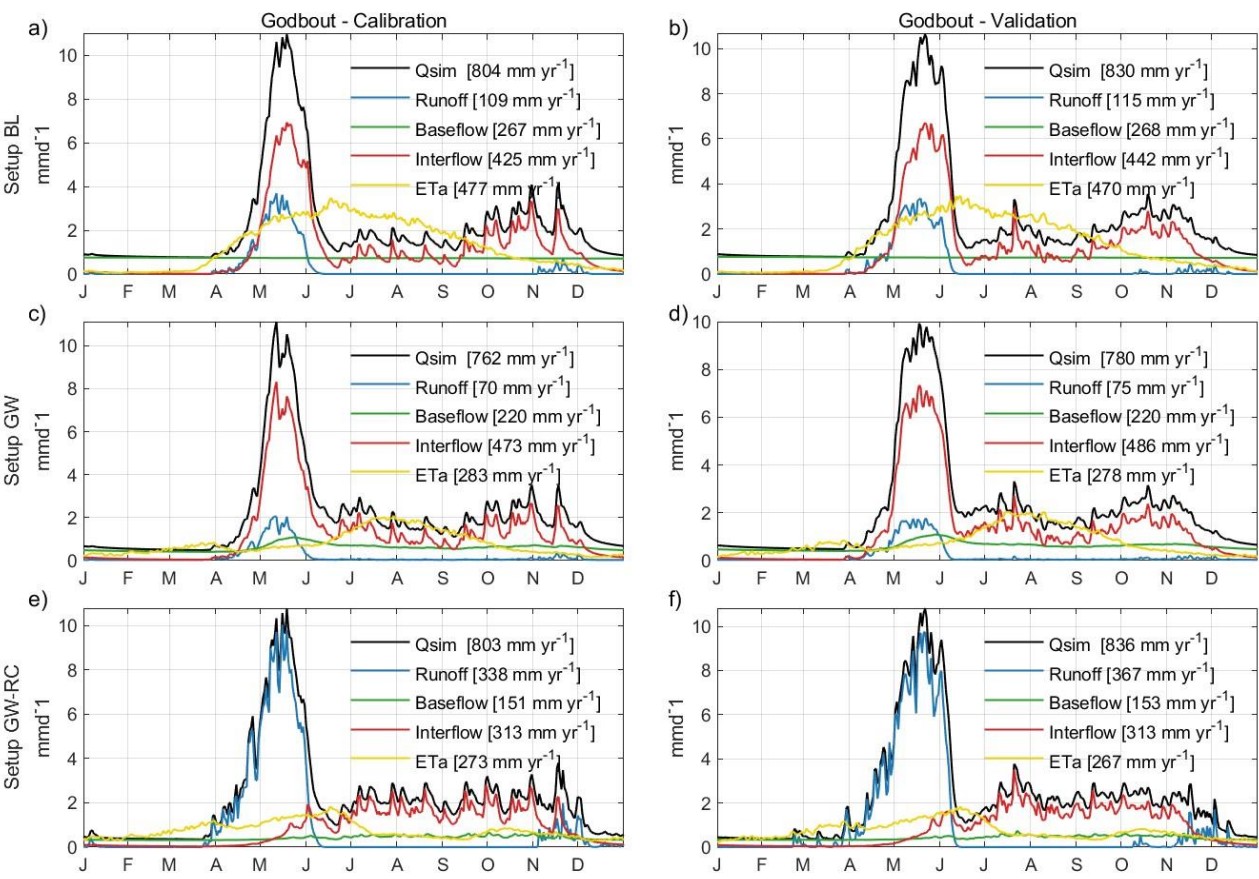

Figure B1. Detailed hydrological variable hydrograph for Godbout catchment during both the calibration and validation phases and for the three configurations. Calibration results are shown in panels (a), (c), and (e) for Configurations BL, GW, and GW-RC, respectively, while validation results are depicted in panels (b), (d), and (f). These hydrographs demonstrate

how baseflow, interflow and runoff contribute to total streamflow throughout the year, with noted annual totals provided for a comprehensive comparison.





## Appendix C

Figure C1. Spatial distribution of annual total recharge for PACES data and hydrological model calibration configurations.


## Code and data availability

The calibrated WaSiM model for all configurations discussed in this study is publicly accessible at https://osf.io/h9rsj/ (Talbot et al., 2024). This dataset encompasses control files, input parameters and output files from both calibration and validation phases.

## Author contribution


FT, JDS and RA conceptualized the study's methodology and designed the model configurations. FT carried out the model simulations, data collection and analysis, interpretation of results and led the writing of the manuscript. JDS and RA supervised the project. AP, GD, JDS, RA and FT discussed the results and contributed to the final manuscript.

## Competing interests

The authors declare that they have no conflict of interest.

## Acknowledgements

This work was funded jointly by the ministère des Ressources naturelles et des Forêts (Quebec, Canada, project number 112332187 conducted at the Direction de la recherche forestière and led by Jean-Daniel Sylvain) and the Forest research service contract number 3322-2022-2187-01 obtained by Richard Arsenault from the Ministère des Ressources naturelles et 705 des Forêts (Quebec, Canada). The authors also acknowledge the use of ChatGPT-4 for assistance in correcting spelling mistakes and improving the flow of text during the manuscript preparation process. The base map in Fig. 1 was created using ArcGIS® software by Esri. ArcGIS® and ArcMap™ are the intellectual property of Esri and are used herein under license. Copyright © Esri. All rights reserved. For more information about Esri® software, please visit www.esri.com.

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
