# Peer review of "Enhancing physically based and distributed hydrological model calibration through internal state variable constraints"

_EGUsphere, 2024_

## Author Comment (AC2)

**Enhancing physically based and distributed hydrological model calibration through internal state variable constraints**

Frédéric Talbot[1], Jean-Daniel Sylvain[2], Guillaume Drolet[2], Annie Poulin[1], Richard Arsenault[1]

[1] Hydrology, Climate and Climate Change Laboratory, École de technologie supérieure, Université du Québec, Montréal, H3C 1K3, Canada

[2] Direction de la recherche forestière, Ministère des Ressources naturelles et des Forêts, Québec, G1P 3W8, Canada

*Correspondence to*: Frédéric Talbot (frederic.talbot.2@ens.etsmtl.ca)

**RC2 (https://doi.org/10.5194/egusphere-2024-3353-RC2)**

This manuscript titled "Enhancing physically based and distributed hydrological model calibration through internal state variable constraints" investigates the effectiveness of various calibration approaches within the Water Balance Simulation Model (WaSiM) to enhance the representation of hydrological variables. The study assesses three configurations: Baseline (BL), Physical Groundwater Model (GW), and Physical Groundwater with Recharge Calibration (GW-RC), which has an add-in of recharge calibration across 34 catchments in Southern Quebec, Canada. The research provides valuable insights into the importance of multi-variable calibration frameworks in developing robust models capable of adapting to anticipated hydrological shifts due to climate change. However, it is too long!

Dear Dr. Modiri,

We thank the reviewer for the thorough evaluation and insightful feedback on our manuscript. The comments are invaluable in refining our paper and we are committed to enhancing its clarity and impact. As for the length of the paper, please see our suggestions below as how we propose to shorten it.

**Major Comments:**

- Abstract:

While the abstract effectively conveys the general research objective and findings, it lacks specific quantitative data. It relies heavily on vague terms and subjective assessments, making it difficult for readers to grasp the magnitude and significance of the improvements achieved fully.

Generally, I suggest you revise it.

We appreciate the critique concerning the lack of specific quantitative data in our abstract, which could limit the reader's understanding of the improvements and findings. Acknowledging this, and in response to both your major and minor comments, we will undertake a thorough revision of the abstract to add quantifiable data that underscore the scope and impact of our research findings. Specifically, we will:

1. Eliminate redundancy, such as the repeated phrase "on the representation of hydrological variables,", on lines 9 and 11.

2. Clarify the study's objectives to provide clear direction regarding the purpose of our research.

3. Include a concise summary of the key findings.

4. Remove vague terms and replace them with specific and measurable outcomes from our study.

5. Include quantifiable results to support the key findings and the study's conclusion.

Methodology:

The authors should provide more details on the selection criteria for the 34 catchments used in the study. While some information is given in section 2.1, a more comprehensive explanation of why these specific catchments were chosen would strengthen the methodology.

This is a good point. We acknowledge the need for a more detailed explanation in Section 2.1 of our manuscript. The catchments were selected based on several criteria to ensure the integrity of the hydrological processes studied: they are free from dams and reservoirs, located away from large urban areas to maintain natural hydrological conditions, and each has a hydrometric station with comprehensive data from 1981 to 2010 for robust model calibration and validation. We also aimed for geographic diversity to cover various climatic conditions across Quebec and, where possible, included catchments covered by the PACES project for data consistency. We plan to revise section 2.1 to provide clarity on our methodological choices.

The selected basins are relatively medium-sized (between 100 and 10,000 square kilometers), which may limit the generalizability of the findings to larger or smaller basins.

We acknowledge the concern regarding the potential limitations in generalizability to very large or very small basins. The chosen size range, from 100 to 10,000 square kilometers, encompasses a broad spectrum that represents a significant portion of catchments typically analyzed in regional hydrological studies. To address this limitation, we will include a discussion on the implications of catchment size in the limitations section of our paper, clarifying the scope of applicability of our findings and suggesting directions for future research that might explore the model's performance across differently sized catchments.

The rationale behind using ERA5 reanalysis data instead of ground-based observations for meteorological inputs should be further elaborated. Since ERA5 recorded underestimating winter precipitation and bias in convective precipitation, it would be better to employ another dataset.

We acknowledge the concerns regarding ERA5's underestimation of winter precipitation and biases in convective precipitation. However, research, such as the study by Tarek et al. (2020), has demonstrated that ERA5-driven hydrological simulations perform comparably to those driven by observational data across Eastern Canada. This study (available at https://doi.org/10.5194/hess-24-2527-2020) showed that, for a broad set of 3138 North American catchments, the results using

ERA5 were equivalent to those using traditional meteorological observations in terms of hydrological modeling accuracy over Eastern Canada.

Additionally, we opted for ERA5 because it offers comprehensive spatial coverage that can be particularly advantageous in regions with sparse weather station networks. While it is true that meteorological stations have their biases, which could introduce different limitations, the uniform coverage of reanalysis data like ERA5 provides a consistent baseline for our study.

Given these points, we justify our preference for ERA5 while recognizing that exploring the impacts of using different meteorological datasets could be a valuable avenue for further research. This point will be expanded in our methodology section to better articulate the rationale behind our choice.

- Model Configurations:

While the three configurations (BL, GW, GW-RC) are described, readers would benefit from a more detailed explanation of
how they differ in their treatment of groundwater processes. The authors should consider discussing the potential limitations of each configuration and how these might impact the results.

We acknowledge that the distinctions and potential limitations among the three model configurations (BL, GW, GW-RC) were not adequately detailed, particularly in their treatment of groundwater processes. In response to your comments, we will revise Section 2.4 of our manuscript to include a clearer comparison of these configurations. Specifically, we will enhance the
description of how each configuration manages groundwater processes, and we will discuss the implications of these methodologies on the results.

- Calibration and Validation:

The split-sample approach for calibration and validation is appropriate, but the authors should discuss any potential impacts of climate non-stationarity on this approach, given the study's focus on climate change adaptation.
We appreciate the comment on the need for a detailed discussion on the potential impacts of climate non-stationarity on our split-sample approach for calibration and validation. We will include a cautionary note regarding the potential limitations due to climate non-stationarity in section 4.4, acknowledging that despite our efforts, the model might not be as robust as anticipated under varying climate conditions. This addition will help clarify the implications and limitations of our approach in the context of climate change adaptation.
Given that you modified the lower and upper boundaries of the model parameter by 10% (L310), a direct comparison with the calibration results of the BL configuration using default parameters might not be entirely fair. Simulating WaSim for all configurations using the adjusted parameter range would be beneficial to ensure a more consistent evaluation.

We understand this comment. To clarify, the GW-RC configuration employs a two-step calibration process. First, it initially uses groundwater recharge data to constrain the parameter range. This begins with a pre-calibration phase applying an objective
function that gives more weight to groundwater recharge metrics—20% for the standard deviation of recharge and 10% for mean annual recharge. Second, after determining a set of parameters from this pre-calibration, we adjust these values by ±10% to define a new, narrower parameter range for the final calibration. This adjustment is applied only to 5 of the 17 calibration parameters. We also make sure that the new parameter range remains within the original bounds. If a boundary exceeds the original range, we adjust it to maintain the parameter within its initial limits. The parameter constraints are based on
groundwater recharge values. Since configurations BL and GW do not integrate recharge in their calibration, they cannot utilize the constrained parameter range as described in this study.

Employing the same constrained parameter range across all configurations would mask the specific impact of including recharge in the calibration, as it would diminish the ability to distinctly evaluate the benefits of this approach. Furthermore, the developed method reduces the degrees of freedom of the GW-RC model, and as such, it is penalized compared to the other models, and thus the results obtained are conservative. If we also shrink the parameter range for the BL model, for example, then the streamflow score can only be reduced as the obtained parameter set would be a subset of the parameter space of the original model. Perhaps the processes would be better represented, however it would not be possible to estimate these bounds without using the recharge data, and thus it is not possible to implement this model in a fair manner.

To enhance clarity and provide a comprehensive understanding, we will expand the description of this calibration methodology in Section 2.4.3 of the manuscript, ensuring a detailed explanation of the process.

Given that the manuscript focuses on WaSim performance, including the computational cost and time associated with each configuration is crucial. This information will be highly valuable for other researchers, allowing them to estimate the resource investment required to achieve comparable improvements in water balance closure.

We recognize the importance of detailing the computational resources required for each configuration of the WaSiM model as highlighted in the comment. Accordingly, we will enhance Table 5 to incorporate the computational demand for each configuration, providing clarity on computational cost in CPU-years (totalling 35 CPU-years on 4.5 GHz CPUs, for your benefit).

**Table 5. Summary of configurations**

| Settings | BL | GW | GW-RC |
|---|---|---|---|
| Groundwater Modelling | Conceptual within unsaturated zone sub-model | Physically based within the groundwater sub-model | Physically based within the groundwater sub-model |
| Calibration Parameters | 17 parameters (including $K_B$ and $Q_0$) | 17 parameters (including Kol and $K_{XY}$) | 17 parameters (including Kol and $K_{XY}$) |
| Precalibration | N/A | N/A | 200 simulations at 1000 meters followed by 50 simulations at 250 meters |
| Calibration | 1000 simulations at 1000 meters followed by 50 simulations at 250 meters | 1000 simulations at 1000 meters followed by 50 simulations at 250 meters | 1000 simulations at 1000 meters followed by 50 simulations at 250 meters |
| Objective function | Kling-Gupta efficiency | Kling-Gupta efficiency | Constrained Kling-Gupta efficiency |
| Computational demand | 10 CPU-year at 4.5 GHz | 10 CPU-year at 4.5 GHz | 15 CPU-year at 4.5 GHz |

CPU-year : A CPU-year is the effort of a CPU running for one year.

- Results Presentation:

Figure 4 highlights the significant shift in the proportion of surface runoff and interflow. Please elaborate on the specific factors that influenced this shift during calibration, particularly considering the inclusion of groundwater recharge in the model.

To address this comment, we will add a sentence in the results section near figure 4 by explicitly stating that the specific factors influencing these shifts, particularly the role of groundwater recharge during calibration, are comprehensively discussed in the discussion section of the manuscript.

The results presented in Figures 3-10 are generally clear, but some figures (e.g., Figures 5, 6, 7) could benefit from additional explanation in the text to help readers interpret the complex information presented.

We agree with this point. In response, we will conduct a thorough review of the text in the results section to ensure that it effectively communicates the major points and provides clearer guidance on interpreting the data presented in these figures.

A more in-depth discussion of the spatial variability in model performance across the 34 catchments would enhance the study's insights, especially when compared with PACES.

This comment highlights a valuable aspect that can indeed enhance the study's insights significantly. We propose to add a few sentences in the discussion to address spatial variability across the 34 catchments. This section will explore the performance variations, providing a deeper understanding of the model's effectiveness in various hydrological settings.

My understanding differs from your conclusion in Figure 10. None of the configurations are aligned with PACES, except for a case in Noire. I would say that the lowest difference is between GW-RC and PACES. In general, I found PACES recharge different than the applied three configurations in this research, according to Figure C1.

We appreciate the observations regarding Figure 10. You're correct in noting the discrepancies between the model configurations and PACES. We will revise the relevant text to ensure it clearly states that the GW-RC configuration provides
"the lowest difference" with PACES instead of "align more closely".

- Climate Change Implications:

While the study mentions the importance of the findings for climate change adaptation, a more specific discussion on how the improved model configurations might be applied in climate change impact assessments would strengthen the paper's relevance.

We thank the reviewer for this suggestion. In response, as stated above, we will expand the discussion section of our manuscript
to emphasize how the refined model configurations aim to improve hydrological process representation, thereby enhancing model robustness in the face of climate change. Additionally, we will incorporate a cautionary note about the potential limitations of our models due to climate non-stationarity. This will help clarify the expected robustness of our models under varying climatic conditions and outline the broader implications for climate change adaptation.

**Minor Comments:**

- Abstract:
    - It exhibits some redundancy, such as the repetition of "on the representation of hydrological variables" in lines 9 and 11.
- The abstract could benefit from a more precise statement of the study's objectives and a more concise summary of the key findings

- Vague Language:
    - "significantly refines the model's ability to depict subsurface processes"
    - "minimal emphasis on recharge"
    - "small and targeted calibration adjustments"
    - "marked improvement"

o   "enhancing the precision"

- Lack of Quantifiable Results:
  - o No specific metrics are mentioned (to quantify the improvement in model performance.
  - o No specific values are given for the improvement in groundwater recharge representation.
  - o No indication of how the "minimal emphasis" on recharge was defined or quantified.

As mentioned in our response to the major comments, we will thoroughly revise the abstract to address the highlighted issues.

- Methodology:
  - o Figure1: Visualising the selected case studies within a coarser-level basin delineation would be beneficial. This would provide context, as the presence of a river traversing the study area can significantly influence catchment behaviour.

To address the suggestion, we propose to revise Figure 1 to add a new panel zoomed in on the Matane catchment to provide more detailed information about the case study.

  - o Table 5: Table's style is totally different from the other presented tables.

    We will adjust the formatting of Table 5 to align with the styling of the other tables presented in the manuscript and to journal standards.

o Could you elaborate on the rationale behind conducting 1000 simulations at 1000 m resolution and only 50 at 250 m resolution? What factors influenced the selection of these specific numbers?

    This is a good question. Our decision to conduct 1000 simulations at 1000 m resolution and only 50 at 250 m resolution was based on preliminary testing on catchments Bonaventure and Matane, which demonstrated that this configuration provides the best balance between computational cost and result accuracy. We tested

75 and 100 simulations at 250 m resolution, but the results were comparable to those obtained with 50 simulations, making the additional computational expense unjustified.

    The key reason we could limit the 250 m simulations to 50 runs is that the calibrated parameters from the 1000 m simulations transferred effectively to the finer resolution, requiring only a slight refinement. We hope this clarifies our approach, and it will be added to the revised version of the paper.

o Given the widespread familiarity of the KGE metric within the research community, a detailed definition in section 2.5.1 may be redundant.

    We agree that a detailed definition of the KGE metric in Section 2.5.1 may be redundant given its widespread familiarity within the research community. To address this, we propose removing the detailed definition and merging Sections 2.5.1 and 2.5.2. This will eliminate redundancy and contribute to a more concise manuscript.

  - o Furthermore, as per comment CC1 received on November 23rd, the assigned weights in section 2.5.2 (L349) require more comprehensive scientific justification and supporting literature.

This is a valid point. Since this methodological step is novel, there is no specific literature directly supporting the assigned weights. To determine these values, we conducted multiple tests with various weight combinations on two test catchments. Our results showed that assigning 20% to the standard deviation of recharge and 10% to the mean recharge provided the best trade-off, ensuring recharge values remained realistic while maintaining acceptable KGE scores. For calibration, a weight of 4% on the recharge standard deviation was sufficient to preserve adequate recharge estimates while achieving strong KGE values, basically providing an incentive to ensure proper process representation without sacrificing too much performance on the streamflow simulation. This will be explained in the text.

o   The manuscript would benefit from considering alternative objective functions besides KGE for streamflow. As suggested in this paper (https://gmd.copernicus.org/articles/11/1873/2018/), using SPAtial EFficiency (SPAEF) could enable the evaluation of multiple hydrological components when you utilise distributed hydrological models. This would provide a more comprehensive assessment of model performance.

This is a good point. We did not use a spatial objective function like SPAEF because we lacked sufficient spatially distributed observations to properly calibrate the model. Applying SPAEF could be an interesting avenue for future studies, particularly when using remote sensing data for calibration. A sentence will be added to section 4.4 of the discussion.

However, to provide a more comprehensive evaluation of model performance across multiple metrics, we propose adding a table in Appendix A (Table A2) that presents calibration and validation results for different configurations. This table includes several key performance metrics beyond KGE, allowing for a broader assessment of model performance. See Table A2 below.

Table A2. Multiple metrics values during calibration and validation periods, for the three configurations.

| Metric | | Calibration | | | Validation | | |
|---|---|---|---|---|---|---|---|
| | | BL | GW | GW-RC | BL | GW | GW-RC |
| KGE | μ | 0.852 | 0.852 | 0.799 | 0.816 | 0.820 | 0.772 |
| | σ | 0.034 | 0.036 | 0.050 | 0.055 | 0.049 | 0.056 |
| Pearson Coefficient | μ | 0.855 | 0.855 | 0.804 | 0.844 | 0.845 | 0.797 |
| | σ | 0.034 | 0.036 | 0.050 | 0.040 | 0.039 | 0.049 |
| Bias ratio | μ | 0.998 | 0.990 | 0.985 | 1.030 | 1.024 | 1.018 |
| | σ | 0.021 | 0.024 | 0.023 | 0.054 | 0.052 | 0.050 |
| Variability ratio | μ | 0.996 | 1.005 | 1.020 | 1.022 | 1.028 | 1.055 |
| | σ | 0.023 | 0.013 | 0.027 | 0.083 | 0.075 | 0.079 |
| NSE | μ | 0.704 | 0.706 | 0.603 | 0.677 | 0.679 | 0.558 |
| | σ | 0.059 | 0.076 | 0.091 | 0.091 | 0.073 | 0.120 |
| RMSE | μ | 20.926 | 20.749 | 24.317 | 22.877 | 22.621 | 26.545 |
| | σ | 11.018 | 10.681 | 13.055 | 12.594 | 11.665 | 13.870 |
| Percent bias | μ | 0.155 | 0.074 | 1.263 | -3.563 | -2.760 | -1.756 |
| | σ | 1.399 | 2.209 | 2.132 | 5.792 | 5.628 | 5.650 |
| MAE | μ | 11.364 | 11.198 | 13.287 | 12.444 | 12.045 | 14.432 |
| | σ | 6.621 | 6.259 | 8.106 | 7.966 | 7.043 | 8.638 |

- Results

  o I would like to know if the same results would be obtained by switching the calibration and validation periods, as indicated in Figure 2. Given that the KGE values for all three setups are relatively close, I am uncertain about the potential benefits of using GW.

  This is a valid concern. We do not have simulations for switched calibration and validation periods, as this would require recalibrating the entire project, which would take several months of computation on our compute infrastructure (see computation time in revised table 5 above). However, to minimize the risk of overfitting to a specific time period, we used an ensemble of 34 catchments distributed across Southern Quebec. Given the diversity of catchments and the three different configurations tested, it is reasonable to hypothesize that the results would remain similar if the calibration and validation periods were reversed, as the likelihood of all configurations overfitting to a particular time period is minimal. Also, the fact that the calibration and validation scores are similar indicates that the models were not overfitted and that expected errors are a good proxy of the generalization error (Hastie et al., 2009) (https://doi.org/10.1007/b94608), and can be used directly, as described in Arsenault et al. (2018). (https://doi.org/10.1016/j.jhydrol.2018.09.027)

o   In Figure3, consider adding each variable's total mean or sum of observations to enhance the visual comparison. This will allow readers to contextualise the calibration and validation boxplots by providing a
reference point for the overall data distribution.

This is a good suggestion. We have modified Figure 3 accordingly. Specifically, we have enhanced the
visualization by adding a wider median line for each boxplot and incorporating a grid in each subplot to
facilitate comparison between periods and configurations. These adjustments improve readability. Please see the updated figure below.

[Figure]

**Figure 3. Boxplots illustrating annual totals (means for groundwater level and soil moisture) variability of model internal variables. These boxplots detail the variability of key hydrological variables modeled with the different configurations, for calibration and validation periods and for all catchments.**

○ The manuscript should provide an explanation for the lack of differentiation in baseflow between configurations GW and GW-RC, as noted in L430

This is a relevant point. While the baseflow of configurations GW and GW-RC appears similar, the differences are statistically significant. This outcome is expected, as both configurations use the same groundwater module, with GW-RC differing only in its calibration method, which accounts for the small
variations observed. However, when compared to BL, both configurations exhibit similar baseflow behavior, indicating that the choice of model configuration primarily drives the differences in baseflow across the three setups. To clarify this, we propose adding a brief explanation at L430.

  ○ Figure 5 reports a difference of around 200 mm/y across all variables among the three configurations for all
34 catchments. To facilitate water balance closure assessment, consider adding a subplot for precipitation data for each basin.

This is a good point. We have modified Figure 5 to include precipitation data in the last subplot to facilitate the assessment of water balance closure. However, since all three configurations use the same precipitation data, the values remain identical across configurations. A revised version of Figure 5 is provided below.

[Figure]

**Figure 5. Boxplots of annual values for key hydrological variables predicted by WaSiM for the 34 catchments and three configurations.**

- The current explanation of Figure 6 was neither informative nor relevant to my perspective. It needs to emphasise the significance of the figure.

  This is a valid concern. While we recognize that it may not be of interest to all readers, we believe it is a valuable addition, as it directly links model parameters to hydrological processes, offering essential insights for WaSiM users. In order to maintain a concise article, we propose moving Figure 6 to the Appendix. This approach allows us to reduce the length of the main text and ensure that readers who are particularly interested in these modeling details can still access the figure.

- Figure7, is the x-axis long-term mean of Q, or are they for a given year? The problem is between October to December in validation period. In the rest, I see no significant differences. Maybe you could drop this figure.

  Figure 7 presents the mean annual hydrographs for the calibration and validation periods. As noted, all three configurations show discrepancies with observed streamflow between October and December during the validation period. Given that the figure does not provide significant additional insights and the manuscript is already lengthy, it will be removed to streamline the article.

  Also, since you have gaps in some of the frozen months (L130), how did you consider them in the likely monthly discharge time series?

  This is a good point. Not all catchments had missing data during the frozen months. To minimize the impact of missing data, we selected calibration and validation years to ensure most catchments had complete records. With this approach, only three out of the 34 catchments had missing data. For these specific catchments, we adjusted the calibration and validation periods to focus on years without gaps. As a result, the final analysis does not include years with missing data. A sentence to this effect will be added at L266 to clarify.

- Language and Style:
  - The manuscript is generally well-written, but there are occasional instances of complex sentence structures that could be simplified for clarity. Thank you for acknowledging the use of ChatGPT-4. The presence of long sentences with numerous commas can be indicative of revised text by an LLM-AI.

    This is a fair point. A thorough verification will be conducted to ensure the text remains clear and concise, and we will have the paper revised by native English speakers to ensure the syntax is less "LLM-y"

- Conclusion:
  - I remain uncertain about the meaning of lines 659-660.

    Line 659-660 will be revised to enhance clarity. The original sentence: *"leads to a more accurate representation of hydrological variables."*

    Will be modified to: *"leads to a representation of hydrological processes that better aligns with expected system behavior."*

This study contributes to hydrological modelling by demonstrating the importance of incorporating internal state variables, particularly groundwater recharge, into model calibration. The authors designed their model well and developed it to have three configurations and further calibrations.

The findings highlight the potential for improved representation of hydrological processes, which is crucial for water resource management and climate adaptation strategies. However, addressing the major and minor comments outlined above would further strengthen the manuscript and enhance its impact on the scientific community. Overall, with appropriate revisions, this paper has the potential to be an important addition to the literature on hydrological model calibration and process representation.

Regarding the initial point about the manuscript length, we propose the following adjustments based on the above responses. Given the widespread familiarity with the KGE metric within the research community, we suggest omitting its definition. Additionally, we will relocate Figure 6 to the appendix and remove Figure 7 from the manuscript. These modifications will result in the removal of two figures and a reduction of approximately 850 words, bringing the manuscript from 9050 words to roughly 8200 words, excluding tables, figure captions, references, appendix, and abstract.

We are eager to implement these changes and believe they will significantly strengthen the manuscript. We thank the reviewer once again for the constructive review, it is much appreciated.

Sincerely,

Frédéric Talbot, on behalf of all authors

---

## Author Response (AR1)

**Enhancing physically based and distributed hydrological model calibration through internal state variable constraints**

Frédéric Talbot1, Jean-Daniel Sylvain2, Guillaume Drolet2, Annie Poulin1, Richard Arsenault1

- ¹ Hydrology, Climate and Climate Change Laboratory, École de technologie supérieure, Université du Québec, Montréal, H3C 1K3, Canada
- 2 Direction de la recherche forestière, Ministère des Ressources naturelles et des Forêts, Québec, G1P 3W8, Canada

Correspondence to: Frédéric Talbot (frederic.talbot.2@ens.etsmtl.ca)

**RC1 (https://doi.org/10.5194/egusphere-2024-3353-RC1)**

Review of "Enhancing Physically Based and Distributed Hydrological Model Calibration through Internal State Variable Constraints"

**General Comments**

15

The authors have presented a comprehensive study that investigates the impact of different calibration approaches on hydrological models using the WaSiM model. The paper explores three distinct configurations: Baseline (BL), Physical Groundwater Model (GW), and Physical Groundwater with Recharge Calibration (GW-RC) to evaluate their effectiveness in representing various hydrological variables. This research addresses an important topic in hydrological modeling by highlighting the significance of integrating internal state variables into the calibration process.

The paper is well-structured, and the authors have made a significant effort to present detailed analyses across multiple catchments. The inclusion of groundwater recharge as a calibration variable is an important approach that aligns with the growing need for multi-variable calibration frameworks in hydrological modeling. The findings make effort to underscore the importance of considering both streamflow and internal hydrological processes for robust model performance.

We thank the reviewer for the thoughtful comments and suggestions regarding our manuscript. The feedback will certainly help improve the clarity and impact of our research.

However, I have a major concern regarding the primary objective of the study, which requires clarification. The current presentation leaves the reader uncertain about whether the study aims to compare calibration strategies or assess the impact of model complexity on hydrological process representation. Addressing this ambiguity will make clear the paper's overall contribution and impact.

We acknowledge the importance of distinctly defining our research focus. To clarify the primary aim of the study, L96-102 was changed from:

"In this study, we implement three distinct model configurations of the WaSiM hydrological model, configuration BL (baseline model), configuration GW (physical groundwater model), and configuration GW-RC (physical groundwater and recharge

calibration model)—to investigate how integrating additional hydrological variables and different calibration approaches influence the representation of hydrological processes over a set of 34 catchments in Nordic conditions."

To:

"In this study, we implement three distinct model configurations of the WaSiM hydrological model: Baseline (BL), which follows a traditional streamflow-based calibration; Physical Groundwater Model (GW), which introduces physically based groundwater flow processes; and Physical Groundwater with Recharge Calibration (GW-RC), which further constrains groundwater recharge during calibration. The objective is to investigate how different calibration strategies and levels of model complexity influence the representation of hydrological processes over a set of 34 catchments in snowy catchment conditions."

**40 Major Comments**

50

55

- **1. Unclear Research Focus**: The primary research question of the paper is not clearly defined. It remains ambiguous whether the authors aim to compare calibration strategies or demonstrate the added value of increasing model complexity.
  - If the goal is to compare calibration strategies, the authors should focus on showing how the constrained recharge parameter improves the realism of the results when compared to both streamflow and PACES data.
- If the goal is to assess model complexity, the paper should clearly outline what unique complexities are introduced in each configuration and how they enhance the model's capability to represent hydrological processes.

We indeed addressed both aspects: comparing calibration strategies and assessing model complexity. Specifically, the manuscript compares the calibration strategies by contrasting the GW and GW-RC configurations and evaluates the model complexity by comparing the BL and GW setups. We acknowledge that this dual focus may not have been articulated clearly enough in the initial draft.

To rectify this, as stated above, we revised the text at L96-102 of the introduction to clearly define and differentiate these two intertwined aims. This clarification will ensure that readers fully understand the scope and the dual objectives of our study.

- **2. Simplifying the Experimental Setup**: The current experimental design includes three configurations (BL, GW, GW-RC), but most of the observed differences in results seem to be attributed to the choice of model complexity rather than calibration strategies.
  - To demonstrate the impact of the constrained recharge parameter, the authors could simplify their experimental setup by comparing two GW-RC experiments: one calibrated solely to streamflow using the KGE metric and another using a modified objective function that accounts for both the mean and variability of recharge. This would directly illustrate the benefit of including internal state variables in the calibration process.
- We understand the suggestion to simplify the experimental setup to more distinctly demonstrate the impact of the constrained recharge parameter. However, our study design, which includes the GW and GW-RC configurations, is specifically crafted to assess the effects of introducing recharge in addition to streamflow in model calibration.

To clarify, both GW and GW-RC configurations operate under the same model complexity, with GW calibrated on streamflow only and GW-RC utilizing both streamflow and recharge during calibration. This setup is intended to explicitly isolate and compare the impact of including recharge alongside streamflow in the calibration process.

To improve the distinction between configurations GW and GW-RC, this sentence was added at the beginning of section 2.4.3 at L311.

"Importantly, GW-RC uses the same model structure as GW, with the goal of isolating the effect of adding groundwater recharge in calibration."

3. Clarifying the Role of GW-RC: The GW-RC configuration is described as integrating recharge calibration into the model. However, the results suggest that most of the observed improvements are due to the activation of more physically based processes rather than the calibration strategy itself. If the authors wish to emphasize the importance of incorporating internal state variables, they should isolate and highlight the specific impact of the recharge constraint.

We acknowledge the importance of clearly distinguishing the impacts attributable to the introduction of recharge in the calibration process.

In the manuscript, the GW and GW-RC configurations operate under the same model complexity, ensuring that any observed improvements can be associated with the calibration strategy rather than model complexity enhancements. The GW configuration is calibrated solely on streamflow metrics, while the GW-RC setup integrates recharge, allowing us to directly compare the effects of adding recharge calibration on model performance.

80 To address your feedback, as stated in the above comment, we added a sentence in the beginning of section 2.4.3.

**Specific Suggestions**

85

95

 Objective Statement: In the introduction, clearly state whether the study aims to evaluate calibration strategies or model complexity.

As stated above, we revised the text at L96-102 of the introduction to clearly define and differentiate these two intertwined aims.

2. **Experimental Design**: Consider restructuring the experimental setup to compare GW-RC configurations with and without recharge constraints. This would make the study's focus more precise.

We appreciate the constructive feedback and would like to clarify some points. As stated above, our experimental design already includes the GW and GW-RC configurations, which are fundamentally identical in terms of model complexity but differ in their calibration targets. The GW configuration is calibrated solely using streamflow data, whereas the GW-RC configuration additionally incorporates recharge data into the calibration process. This setup precisely allows us to isolate and compare the effects of integrating recharge in calibration. In response to your suggestion, we enhanced the clarity of this aspect in the methodology section. As stated above, we added a sentence in the beginning of section 2.4.3 to address this concern.

3. **Discussion Section**: Emphasize the role of model complexity in the observed differences in results. If the paper aims to introduce new complexities in the model, demonstrate their unique contribution to the model's performance.

We appreciate the input on the need to better highlighting the distinct contributions of each model configuration to the observed results. To address the comment, we revised the text at L300-304 to better articulate how the baseline (BL) configuration, utilizing a conceptual method for groundwater flow, contrasts with the GW configuration which employs a physically based approach to groundwater dynamics. Here is the added explanation:

- "In WaSiM, the groundwater model is coupled bi-directionally with the unsaturated zone, ensuring a dynamic exchange of water fluxes. The unsaturated zone module calculates fluxes between the unsaturated zone and the groundwater that act as the upper boundary condition for the groundwater model, while the groundwater module simulates lateral flow and adjusts the groundwater table, feeding back changes to the unsaturated zone as inflow or outflow."
  - 4. **Results Interpretation**: Clearly distinguish between the impact of calibration strategies and model complexity in the results section. For example, highlight how much of the improvement in GW-RC is due to the activation of groundwater dynamics versus the recharge calibration constraint.

We acknowledge this concern. However, we have already addressed this distinction in our article. Our study was designed to differentiate between model complexity and calibration methods using three configurations. When comparing configuration BL and GW, the differences are due to model complexity, as the BL configuration uses a conceptual method for groundwater flow, while GW employs physically based equations. Conversely, when comparing configuration GW and GW-RC, the improvements are due to the integration of groundwater recharge into the calibration process, not model complexity. Both GW and GW-RC configurations employ the same level of physical process modelling, ensuring that any performance differences are due to calibration strategies alone.

**Conclusion**

105

110

120

The paper presents valuable insights into the role of multi-variable calibration in hydrological modeling. However, the main research target needs to be more clearly defined. The authors should clarify whether their focus is on calibration strategy comparison or model complexity assessment. Additionally, simplifying the experimental setup to directly compare the impact of recharge constraints would make the study more impactful.

We believe the changes address the concerns and enhance the manuscript. We thank the reviewer once again for the constructive comments.

Best regards,

Frédéric Talbot on behalf of all authors

**RC2 (https://doi.org/10.5194/egusphere-2024-3353-RC2)**

This manuscript titled "Enhancing physically based and distributed hydrological model calibration through internal state variable constraints" investigates the effectiveness of various calibration approaches within the Water Balance Simulation Model (WaSiM) to enhance the representation of hydrological variables. The study assesses three configurations: Baseline (BL), Physical Groundwater Model (GW), and Physical Groundwater with Recharge Calibration (GW-RC), which has an addin of recharge calibration across 34 catchments in Southern Quebec, Canada. The research provides valuable insights into the

importance of multi-variable calibration frameworks in developing robust models capable of adapting to anticipated hydrological shifts due to climate change. However, it is too long!

Dear Dr. Modiri,

135

140

145

We thank the reviewer for the thorough evaluation and insightful feedback on our manuscript. The comments are invaluable in refining our paper and we are committed to enhancing its clarity and impact. As for the length of the paper, we did the following adjustments to shorten the manuscript. Given the widespread familiarity with the KGE metric within the research community, we remove section 2.5.1 containing its definition. Additionally, Figure 6 was relocated to the appendix and Figure 7 was removed from the manuscript.

Moreover, after carefully reviewing the PACES datasets, we decided that it would be more appropriate not to compare the spatial distribution of our simulated groundwater recharge with the spatial patterns provided by PACES. Although PACES is available at a spatial resolution of 250 meters, its distributed values are derived from large-scale water balance estimates that are subsequently downscaled based on aquifer and soil types. This approach results in a product that is fundamentally coarse in nature and not intended for detailed spatial interpretation at the pixel level. In addition, the methodology used to generate PACES varies across regions, which can lead to abrupt discontinuities in estimated recharge values across administrative boundaries, as observed in certain catchments like Bécancour, which showed a large jump in absolute values at the boundary intersections. These inconsistencies undermine the reliability of PACES as a reference for distributed model evaluation.

However, PACES remains useful for providing long-term average recharge estimates at the catchment scale, which we used during calibration to ensure the internal water balance of our model was physically consistent. For this reason, while we retain the use of PACES for catchment-averaged comparisons, we excluded spatially explicit comparisons from this study to avoid misinterpretation and ensure methodological rigor.

To ensure clarity and methodological consistency, we removed the spatial comparison from the manuscript, which led to the removal of Figure 10, its associated text, and Appendix C. The remaining content of Section 3.4 was merged into Section 3.3 of the results.

In total, these changes resulted in the removal of three figures and two subsections from the manuscript, contributing to our efforts in shortening the paper, which is another point that was raised by reviewers.

**155 Major Comments:**

**Abstract:**

While the abstract effectively conveys the general research objective and findings, it lacks specific quantitative data. It relies heavily on vague terms and subjective assessments, making it difficult for readers to grasp the magnitude and significance of the improvements achieved fully.

160 Generally, I suggest you revise it.

We appreciate the critique concerning the lack of specific quantitative data in our abstract, which could limit the reader's understanding of the improvements and findings. Acknowledging this, and in response to both your major and minor comments, we undertook a thorough revision of the abstract. In the new abstract, we eliminated redundancy, clarified the

study's objective, included a concise summary of the key findings, removed vague terms and included quantifiable results.

165 Here is the new version of the abstract:

"Accurately representing hydrological processes remains a major challenge in hydrological modeling. Recent studies have demonstrated the benefits of multi-variable calibration, which integrates additional hydrological variables such as evapotranspiration and soil moisture alongside streamflow to improve model realism. However, groundwater recharge as a calibration variable remains relatively underexplored.

170 This study evaluates how incorporating groundwater recharge into the calibration of the Water Balance Simulation Model (WaSiM) affects hydrological variables representation. Three configurations were tested: Baseline (BL) with streamflow-only calibration, Physical Groundwater Model (GW) with physically-based groundwater flow, and Physical Groundwater with Recharge Calibration (GW-RC), which further constrains groundwater recharge during calibration. The models were calibrated and applied to 34 catchments in Southern Québec. Their performance was evaluated using the Kling-Gupta 175 Efficiency (KGE) for streamflow and spatial estimates of groundwater recharge derived from a previous research project conducted in the same region.

Results indicate that while calibrating on streamflow alone produces high KGE values (median KGE = 0.83 for GW and 0.82 for BL), but it comes at the cost of misrepresenting subsurface hydrological processes. Adding groundwater recharge constraints (GW-RC) reduce streamflow performance, with a median KGE of 0.77 for GW-RC, but improves hydrological variable representation, especially in seasonal runoff patterns, where it better captures the balance between surface runoff and interflow during snowmelt. Additionally, GW-RC showed the smallest differences with the groundwater recharge estimates.

These findings illustrate the consequence of equifinality in streamflow-based calibration, where multiple parameter sets can yield similar streamflow outputs while misrepresenting internal hydrological processes. Incorporating groundwater recharge constraints improves the representation of internal hydrological processes while maintaining strong streamflow simulation performance, which could ultimately enhance reliability of climate change adaptation and water resource management strategies."

**Methodology:**

180

185

195

The authors should provide more details on the selection criteria for the 34 catchments used in the study. While some information is given in section 2.1, a more comprehensive explanation of why these specific catchments were chosen would strengthen the methodology.

This is a good point. We acknowledge the need for a more detailed explanation in Section 2.1 of our manuscript. The catchments were selected based on several criteria to ensure the integrity of the hydrological processes studied: they are free from dams and reservoirs, located away from large urban areas to maintain natural hydrological conditions, and each has a hydrometric station with comprehensive data from 1981 to 2010 for robust model calibration and validation. We also aimed for geographic diversity to cover various climatic conditions across Quebec and, where possible, included catchments covered

by the PACES project for data consistency. To provide more details on why these specific catchments were chosen, L107-114 was modified in section 2.1. Here is the original text:

"These specific catchments were selected for their inclusion in the Hydroclimatic Atlas of Southern Québec (MDDELCC, 2022) due to the availability of comprehensive streamflow data and their representation of the diverse hydrological conditions prevalent throughout Southern Quebec. Selected catchments are unaffected by the presence of dams and reservoirs, preserving the natural integrity of hydrological processes."

**Here is the new text:**

205

210

215

220

225

"These catchments were selected based on several key criteria to ensure robust model calibration and validation. Specifically, they were selected based on the availability of comprehensive streamflow data from 1981 to 2010. Additionally, catchments were selected to represent the region's geographical and hydrological diversity to capture a range of climatic conditions across the study area. Where possible, catchments covered by the PACES project were prioritized to ensure data consistency and facilitate comparisons of groundwater recharge estimates. To preserve the natural integrity of hydrological processes under study, selected catchments needed to be free from dams and reservoirs and located away from major urban areas to minimize anthropogenic influences."

The selected basins are relatively medium-sized (between 100 and 10,000 square kilometers), which may limit the generalizability of the findings to larger or smaller basins.

We acknowledge the concern regarding the potential limitations in generalizability to very large or very small basins. The chosen size range, from 100 to 10,000 square kilometers, encompasses a broad spectrum that represents a significant portion of catchments typically analyzed in regional hydrological studies. To address the limitations in generalizability of the results due to catchment size, a short paragraph was added at L636 of section 4.4 of the discussion section. Here is the added paragraph:

"Additionally, the selected catchments in this study range from 525 km² to 6,840 km², which may limit the generalizability of the findings to catchments outside this size range. Future research could investigate smaller or larger catchments to determine whether the observed trends and calibration impacts remain consistent across different watershed scales."

The rationale behind using ERA5 reanalysis data instead of ground-based observations for meteorological inputs should be further elaborated. Since ERA5 recorded underestimating winter precipitation and bias in convective precipitation, it would be better to employ another dataset.

We acknowledge the concerns regarding ERA5's underestimation of winter precipitation and biases in convective precipitation. However, research, such as the study by Tarek et al. (2020), has demonstrated that ERA5-driven hydrological simulations perform comparably to those driven by observational data across Eastern Canada. This study (available at <a href="https://doi.org/10.5194/hess-24-2527-2020">https://doi.org/10.5194/hess-24-2527-2020</a>) showed that, for a broad set of 3138 North American catchments, the results using ERA5 were equivalent to those using traditional meteorological observations in terms of hydrological modeling accuracy over Eastern Canada.

Additionally, we opted for ERA5 because it offers comprehensive spatial coverage that can be particularly advantageous in regions with sparse weather station networks. While it is true that meteorological stations have their biases, which could introduce different limitations, the uniform coverage of reanalysis data like ERA5 provides a consistent baseline for our study. Given these points, we justify our preference for ERA5 while recognizing that exploring the impacts of using different meteorological datasets could be a valuable avenue for further research. The choice of using ERA5 reanalysis data was further elaborated at L136-143 in section 2.2.1. Here is the original text:

"These datasets effectively overcome the limitations of observational data and have demonstrated performance on par with observational records in this region (Tarek et al., 2020)."

**Here is the revised text:**

240

245

260

"While ERA5 is known to underestimate winter precipitation and exhibit biases in convective precipitation, studies such as Tarek et al. (2020) have demonstrated that ERA5-driven hydrological simulations perform comparably to those using ground-based observational data across Eastern Canada. Their evaluation of 3138 North American catchments found that ERA5-based simulations achieved similar accuracy levels to traditional meteorological observations in hydrological modeling, particularly in Eastern Canada. While observational data can offer higher local accuracy, it also comes with gaps and inconsistencies due to station distribution and measurement errors. ERA5 provided gridded and consistent meteorological inputs across all study catchments, reducing potential biases from heterogeneous station networks."

**• Model Configurations:**

While the three configurations (BL, GW, GW-RC) are described, readers would benefit from a more detailed explanation of how they differ in their treatment of groundwater processes. The authors should consider discussing the potential limitations of each configuration and how these might impact the results.

We acknowledge that the distinctions and potential limitations among the three model configurations (BL, GW, GW-RC) were not adequately detailed, particularly in their treatment of groundwater processes. To enhance the explanation on how configurations BL and GW differ in their treatment of groundwater processes, an explanation was added at L300 of section 2.4.2. Here is the added explanation:

"In WaSiM, the groundwater model is coupled bi-directionally with the unsaturated zone, ensuring a dynamic exchange of water fluxes. The unsaturated zone module calculates fluxes between the unsaturated zone and the groundwater that act as the upper boundary condition for the groundwater model, while the groundwater module simulates lateral flow and adjusts the groundwater table, feeding back changes to the unsaturated zone as inflow or outflow."

**• Calibration and Validation:**

The split-sample approach for calibration and validation is appropriate, but the authors should discuss any potential impacts of climate non-stationarity on this approach, given the study's focus on climate change adaptation.

We appreciate the comment on the need for a detailed discussion on the potential impacts of climate non-stationarity on our split-sample approach for calibration and validation. A description on the impact of climate non-stationarity and how the refined model configurations enhance model robustness in climate change impact studies was added at L608 of section 4.4.

"By improving the representation of hydrological processes, the GW-RC configuration may enhance the model's ability to simulate hydrological responses under changing climatic conditions. This is especially important given the non-stationarity of climate, where historical hydrological relationships no longer hold under future conditions. In this context, calibrating models using physically meaningful constraints, such as groundwater recharge, may improve their ability to capture shifting hydrological patterns and enhance confidence in assessments of climate change impacts on hydrological variables."

265

275

280

290

295

Given that you modified the lower and upper boundaries of the model parameter by 10% (L310), a direct comparison with the calibration results of the BL configuration using default parameters might not be entirely fair. Simulating WaSim for all configurations using the adjusted parameter range would be beneficial to ensure a more consistent evaluation.

We understand this comment. To clarify, the GW-RC configuration employs a two-step calibration process. First, it initially uses groundwater recharge data to constrain the parameter range. This begins with a pre-calibration phase applying an objective function that gives more weight to groundwater recharge metrics—20% for the standard deviation of recharge and 10% for mean annual recharge. Second, after determining the best set of parameters from this pre-calibration, we adjust these values by  $\pm 10\%$  to define a new, narrower parameter range for the final calibration. This adjustment is applied only to 5 of the 17 calibration parameters. We also make sure that the new parameter range remains within the original bounds. If a boundary exceeds the original range, we adjust it to maintain the parameter within its initial limits. The parameter constraints are based on groundwater recharge values. Since configurations BL and GW do not integrate recharge in their calibration, they cannot utilize the constrained parameter range as described in this study.

Employing the same constrained parameter range across all configurations would mask the specific impact of including recharge in the calibration, as it would diminish the ability to distinctly evaluate the benefits of including ground water recharge. Furthermore, the developed method reduces the degrees of freedom of the GW-RC model, and as such, it is penalized compared to the other models, and thus the results obtained are conservative.

285 To enhance clarity around calibration parameters constraint, this explanation was added at L336 of section 2.4.3:

"A key justification for not applying the same constrained parameter range across all configurations is that BL and GW do not incorporate recharge in calibration. Their parameters optimization is based solely on streamflow, whereas GW-RC explicitly integrates recharge to constrain the parameters range."

Given that the manuscript focuses on WaSim performance, including the computational cost and time associated with each configuration is crucial. This information will be highly valuable for other researchers, allowing them to estimate the resource investment required to achieve comparable improvements in water balance closure.

We recognize the importance of detailing the computational resources required for each configuration of the WaSiM model as highlighted in the comment. Accordingly, we enhanced Table 5 to incorporate the computational demand for each configuration, providing clarity on computational cost in CPU-years (totalling 35 CPU-years on 4.5 GHz CPUs, for your benefit). Here is the revised version of the table:

**Table 5. Summary of configurations**

| Settings                  | BL                                                                             | GW                                                                             | GW-RC                                                                          |  |  |
|---------------------------|--------------------------------------------------------------------------------|--------------------------------------------------------------------------------|--------------------------------------------------------------------------------|--|--|
| Groundwater
Modelling  | Conceptual within unsaturated zone sub-model                                   | Physically based within the groundwater sub-model                              | Physically based within the groundwater sub-model                              |  |  |
| Calibration
Parameters | 17 parameters (including $K_B$ and $Q_0$ )                                     | 17 parameters (including Kol and $K_{XY}$ )                                    | 17 parameters (including Kol and $K_{XY}$ )                                    |  |  |
| Precalibration            | N/A                                                                            | N/A                                                                            | 200 simulations at 1000 meters
followed by 50 simulations at 250
meters  |  |  |
| Calibration               | 1000 simulations at 1000 meters
followed by 50 simulations at 250
meters | 1000 simulations at 1000 meters
followed by 50 simulations at 250
meters | 1000 simulations at 1000 meters
followed by 50 simulations at 250
meters |  |  |
| Objective function        | Kling-Gupta efficiency                                                         | Kling-Gupta efficiency                                                         | Constrained Kling-Gupta efficiency                                             |  |  |
| Computational demand      | 10 CPU-year at 4.5 GHz                                                         | 10 CPU-year at 4.5 GHz                                                         | 15 CPU-year at 4.5 GHz                                                         |  |  |

CPU-year: A CPU-year is the effort of a CPU running for one year.

**• Results Presentation:**

300

310

Figure 4 highlights the significant shift in the proportion of surface runoff and interflow. Please elaborate on the specific factors that influenced this shift during calibration, particularly considering the inclusion of groundwater recharge in the model.

The specific factors influencing these shifts are already described in the discussion section. In response to this comment, we added a sentence in the results section near figure 4 by explicitly stating that the specific factors influencing these shifts are comprehensively discussed in the discussion section of the manuscript. Here is the sentence added at L428:

"The factors influencing the differences between configurations are further analyzed in the discussion section."

The results presented in Figures 3-10 are generally clear, but some figures (e.g., Figures 5, 6, 7) could benefit from additional explanation in the text to help readers interpret the complex information presented.

We agree with this point. Figure 6 was moved to Appendix C. Figure 7 was removed from the article. The text supporting Figure 5 was revised. Here is the original text:

"Figure 5 illustrates the distribution of key hydrological variables for the 34 catchments and for each configuration. Consistent trends in hydrological responses are observed across the catchments for each model configuration. For instance, configuration GW-RC typically shows higher runoff and lower interflow values across most catchments. Similarly, configuration BL consistently reports higher actual evapotranspiration and lower groundwater recharge. These patterns, initially observed in Fig. 3 and Fig. 4, are corroborated across most catchments, aligning with the statistical findings presented in Table 6."

**Here is the revised text:**

"Figure 5 illustrates the annual totals distribution of key hydrological variables (surface runoff, baseflow, interflow, actual evapotranspiration, groundwater recharge, and precipitation) across 34 catchments for each model configuration (BL, GW, and GW-RC). The figure provides a comprehensive comparison of how each configuration partitions the water balance components for each catchment. Consistent trends in hydrological responses are observed across the catchments for each

model configuration. For instance, configuration GW-RC shows higher surface runoff and baseflow, with lower interflow values compared to the other configurations indicating that calibration strategies and model complexity influence the distribution of water fluxes. In contrast, configuration BL consistently reports higher actual evapotranspiration (ETa) and lower groundwater recharge. Statistical comparisons indicated that baseflow, surface runoff and interflow dynamics of GW-RC configuration are significantly different compared to BL and GW configurations (Table 6)."

A more in-depth discussion of the spatial variability in model performance across the 34 catchments would enhance the study's insights, especially when compared with PACES.

This comment highlights a valuable aspect that can indeed enhance the study's insights significantly. A few sentences were added at L565 to discuss the spatial variability across catchments when comparing with the PACES results. Here is the added text:

"The spatial analysis of groundwater recharge across the catchments revealed key differences between the model configurations. Configuration BL struggled to simulate recharge rates exceeding 250 mm yr-1, despite such values being common in the study area. However, it performed well in catchments with low recharge values, consistently producing lower recharge estimates compared to GW and GW-RC.

For configurations GW and GW-RC, groundwater recharge rates were influenced by catchment size and total precipitation. Larger catchments with higher precipitation exhibited greater recharge, while smaller, drier catchments showed lower recharge rates. This relationship indicates that these configurations better capture broad spatial trends in groundwater recharge compared to configuration BL, which showed less sensitivity to variations in precipitation and catchment size. Furthermore, GW and GW-RC displayed similar spatial patterns. Configuration GW exhibited the highest variability between catchments, whereas GW-RC produced estimates of average annual recharge that were more consistent with PACES data across most catchments. Future studies should further investigate how spatial characteristics of catchments affect the overall dynamics of hydrological variables in this context."

My understanding differs from your conclusion in Figure 10. None of the configurations are aligned with PACES, except for a case in Noire. I would say that the lowest difference is between GW-RC and PACES. In general, I found PACES recharge different than the applied three configurations in this research, according to Figure C1.

We appreciate the reviewer's observations regarding the comparison with PACES. Following a careful review, we have decided to remove Figures 10 and C1 from the manuscript, along with the associated discussion, due to concerns about the spatial reliability of the PACES dataset. As noted in our revised text, while PACES remains useful for long-term, catchment-averaged recharge estimates, it is not suitable for spatially distributed comparisons.

**• Climate Change Implications:**

335

340

345

While the study mentions the importance of the findings for climate change adaptation, a more specific discussion on how the improved model configurations might be applied in climate change impact assessments would strengthen the paper's relevance. We thank the reviewer for this suggestion. In response, as stated above, we expanded the discussion section of our manuscript at L608 to emphasize how the refined model configurations aim to improve hydrological process representation, thereby

enhancing model robustness in the face of climate change. Additionally, we incorporated a cautionary note about the potential limitations of our models due to climate non-stationarity.

**355 **Minor Comments:**

360

365

370

375

- Abstract:
- o It exhibits some redundancy, such as the repetition of "on the representation of hydrological variables" in lines 9 and 11.
  - The abstract could benefit from a more precise statement of the study's objectives and a more concise summary of the key findings
  - Vague Language:
    - o "significantly refines the model's ability to depict subsurface processes"
    - o "minimal emphasis on recharge"
    - o "small and targeted calibration adjustments"
    - o "marked improvement"
    - "enhancing the precision"
  - Lack of Quantifiable Results:
    - O No specific metrics are mentioned (to quantify the improvement in model performance.
    - o No specific values are given for the improvement in groundwater recharge representation.
    - o No indication of how the "minimal emphasis" on recharge was defined or quantified.

As mentioned in our response to the major comments, we revised the abstract to address all highlighted issues.

- Methodology:
  - Figure 1: Visualising the selected case studies within a coarser-level basin delineation would be beneficial.
     This would provide context, as the presence of a river traversing the study area can significantly influence catchment behaviour.

Figure 1 was modified, and a new zoomed panel was added to provide more details about the case study catchment.

Figure 1. Elevation map of study catchments in southern Quebec.

385

390

- o Table 5: Table's style is totally different from the other presented tables.
- We adjusted the formatting of Table 5 to align with the styling of the other tables presented in the manuscript and to journal standards.
  - Oculd you elaborate on the rationale behind conducting 1000 simulations at 1000 m resolution and only 50 at 250 m resolution? What factors influenced the selection of these specific numbers?

This is a good question. Our decision to conduct 1000 simulations at 1000 m resolution and only 50 at 250 m resolution was based on preliminary testing on catchments Bonaventure and Matane, which demonstrated that this approach provides the best balance between computational cost and result accuracy. We tested 75 and 100 simulations at 250 m resolution, but the results were comparable to those obtained with 50 simulations, making the additional computational expense unjustified.

The key reason we could limit the 250 m simulations to 50 runs is that the calibrated parameters from the 1000 m simulations transferred effectively to the finer resolution, requiring only a slight refinement. For hilly regions like our study area, the WaSiM documentation (Schulla, 2024 link: http://www.wasim.ch/en/products/wasim\_description.htm) recommends a maximum spatial resolution between 2000 m and 5000 m. By working at a resolution well below this threshold, we maintain model accuracy. We hope this clarifies our approach.

The rationale behind the spatial resolution choice and number of evaluations was added at L263. Here is the added text:

"This two-step approach was chosen based on preliminary testing on the Bonaventure and Matane catchments, which demonstrated that transferring optimized parameters from 1000 m resolution to 250 m required only minor refinements. Additional tests showed that increasing the number of simulations at 250 m resolution beyond 50 runs (e.g., 75 or 100) provided negligible improvements in model performance, making further computational expense unjustified."

- Given the widespread familiarity of the KGE metric within the research community, a detailed definition in section 2.5.1 may be redundant.
- 400 We agree that a detailed definition of the KGE metric in Section 2.5.1 may be redundant given its widespread familiarity within the research community. To address this, we removed the detailed definition and merged Sections 2.5.1 and 2.5.2. This eliminated redundancy and contributed to a more concise manuscript.
  - Furthermore, as per comment CC1 received on November 23rd, the assigned weights in section 2.5.2 (L349)
     require more comprehensive scientific justification and supporting literature.
- 405 This is a valid point. Since this methodological step is novel, there is no specific literature directly supporting the assigned weights. To determine these values, we conducted multiple tests with various weight combinations on two test catchments. Our results showed that assigning 20% to the standard deviation of recharge and 10% to the mean recharge provided the best trade-off, ensuring recharge values remained realistic while maintaining acceptable KGE scores. For calibration, a weight of 4% on the recharge standard deviation was sufficient to preserve adequate recharge estimates while achieving strong KGE values, basically providing an incentive to ensure proper process representation without sacrificing too much performance on the streamflow simulation. An explanation of the assigned weight in the objective function was provided at L349-354 of section 2.5.2. Here is the original text:

"This specific weighting was chosen based on preliminary tests, where various weight combinations were evaluated on a test catchment."

**415 Here is the new version:**

420

"This specific weighting was determined based on preliminary testing conducted on two test catchments, where various weight combinations were evaluated. The selected weights provided the best trade-off, ensuring that recharge estimates remained realistic while maintaining strong KGE values for streamflow. In particular, assigning 20% to the recharge standard deviation and 10% to the mean annual recharge allowed the model to better capture recharge variability without compromising overall streamflow performance."

- The manuscript would benefit from considering alternative objective functions besides KGE for streamflow. As suggested in this paper (https://gmd.copernicus.org/articles/11/1873/2018/), using SPAtial EFficiency (SPAEF) could enable the evaluation of multiple hydrological components when you utilise distributed hydrological models. This would provide a more comprehensive assessment of model performance.
- This is a good point. We did not use a spatial objective function like SPAEF because we lacked sufficient good quality spatially distributed observations to properly calibrate the model. Applying SPAEF could be an interesting avenue for future studies,

particularly when using remote sensing data for calibration. To address the consideration of alternative objective functions, a short paragraph was added at L639 of section 4.4 of the discussion section. Here is the added paragraph:

"Furthermore, the choice of objective function presents another limitation. This study primarily relied on the Kling-Gupta

Efficiency (KGE) for streamflow calibration. However, alternative metrics such as SPAtial Efficiency (SPAEF) (Koch et al.,

2018) could enable a more comprehensive evaluation of multiple hydrological components when using distributed
hydrological models. The lack of sufficient spatially distributed observations prevented the application of SPAEF in this study,
but future research could explore its use, particularly in conjunction with remote sensing data to better assess the spatial
coherence of hydrological variables."

However, to provide a more comprehensive evaluation of model performance across multiple metrics, Table A2 was added to Appendix A to present calibration and validation results on several performance metrics, allowing for a broader assessment of model performance.

Table A2. Multiple streamflow metrics values during calibration (2000-2009) and validation (1990-1999) periods, for the three configurations.

| Matria              |   | Calibration |        | Validation   |        |                        |        |
|---------------------|---|-------------|--------|--------------|--------|------------------------|--------|
| Metric              |   | BL          | GW     | GW-RC | BL     | $\mathbf{G}\mathbf{W}$ | GW-RC  |
| KGE                 | μ | 0.852       | 0.852  | 0.799        | 0.816  | 0.820                  | 0.772  |
| KUE                 | σ | 0.034       | 0.036  | 0.050        | 0.055  | 0.049                  | 0.056  |
| Pearson Coefficient | μ | 0.855       | 0.855  | 0.804        | 0.844  | 0.845                  | 0.797  |
| rearson Coefficient | σ | 0.034       | 0.036  | 0.050        | 0.040  | 0.039                  | 0.049  |
| Discounts           | μ | 0.998       | 0.990  | 0.985        | 1.030  | 1.024                  | 1.018  |
| Bias ratio          | σ | 0.021       | 0.024  | 0.023        | 0.054  | 0.052                  | 0.050  |
| Variability ratio   | μ | 0.996       | 1.005  | 1.020        | 1.022  | 1.028                  | 1.055  |
| variability ratio   | σ | 0.023       | 0.013  | 0.027        | 0.083  | 0.075                  | 0.079  |
| NSE                 | μ | 0.704       | 0.706  | 0.603        | 0.677  | 0.679                  | 0.558  |
| NSE                 | σ | 0.059       | 0.076  | 0.091        | 0.091  | 0.073                  | 0.120  |
| RMSE                | μ | 20.926      | 20.749 | 24.317       | 22.877 | 22.621                 | 26.545 |
| KWSE                | σ | 11.018      | 10.681 | 13.055       | 12.594 | 11.665                 | 13.870 |
| Percent bias        | μ | 0.155       | 0.074  | 1.263        | -3.563 | -2.760                 | -1.756 |
| reicent dias        | σ | 1.399       | 2.209  | 2.132        | 5.792  | 5.628                  | 5.650  |
| MAE                 | μ | 11.364      | 11.198 | 13.287       | 12.444 | 12.045                 | 14.432 |
| IVIAE               | σ | 6.621       | 6.259  | 8.106        | 7.966  | 7.043                  | 8.638  |

Results

440

- O I would like to know if the same results would be obtained by switching the calibration and validation periods, as indicated in Figure 2. Given that the KGE values for all three setups are relatively close, I am uncertain about the potential benefits of using GW.
- This is a valid concern. We do not have simulations for switched calibration and validation periods, as this would require recalibrating the entire project, which would take several months of computation on our compute infrastructure (see computation time in revised table 5 above). However, to minimize the risk of overfitting to a specific time period, we used an ensemble of 34 catchments distributed across Southern Quebec. Given the diversity of catchments and the three different configurations tested, it is reasonable to hypothesize that the results would remain similar if the calibration and validation periods were reversed, as the likelihood of all configurations overfitting to a particular time period is minimal. Also, the fact that the calibration and validation scores are similar indicates that the models were not overfitted and that expected errors are a good proxy of the generalization error (Hastie et al., 2009) (https://doi.org/10.1007/b94608), and can be used directly, as described in Arsenault et al. (2018). (https://doi.org/10.1016/j.jhydrol.2018.09.027)

455

 In Figure3, consider adding each variable's total mean or sum of observations to enhance the visual comparison. This will allow readers to contextualise the calibration and validation boxplots by providing a reference point for the overall data distribution.

This is a good suggestion. We have modified Figure 3 accordingly. Specifically, we have enhanced the visualization by adding a wider median line for each boxplot and incorporating a grid in each subplot to facilitate comparison between periods and configurations. These adjustments improve readability. Please see the updated figure below.

Figure 3. Boxplots illustrating annual totals (means for groundwater level and soil moisture) variability of model internal variables.

These boxplots detail the variability of key hydrological variables modeled with the different configurations, for calibration and validation periods and for all catchments.

 The manuscript should provide an explanation for the lack of differentiation in baseflow between configurations GW and GW-RC, as noted in L430

This is a relevant point. While the baseflow of configurations GW and GW-RC appears similar, the differences are statistically significant. This outcome is expected, as both configurations use the same groundwater module, with GW-RC differing only in its calibration method, which accounts for the small variations observed. However, when compared to BL, both configurations exhibit similar baseflow behavior, indicating that the choice of model configuration primarily drives the differences in baseflow across the three setups. To clarify this, a sentence was added at L437 discussing the differences in baseflow results across the three configurations. Here is the added text:

465

470

"This outcome is expected, as both GW and GW-RC employ the same groundwater module, with GW-RC differing only in its calibration approach. The observed variations in baseflow arise from the inclusion of recharge constraints in GW-RC. More broadly, the significant contrast in baseflow between BL and the other two configurations suggests that the choice of model configuration plays a primary role in determining baseflow dynamics rather than the specific calibration strategy applied."

- Figure 5 reports a difference of around 200 mm/y across all variables among the three configurations for all 34 catchments. To facilitate water balance closure assessment, consider adding a subplot for precipitation data for each basin.
- This is a good point. We have modified Figure 5 to include precipitation data in the last subplot to facilitate the assessment of water balance closure. However, since all three configurations use the same precipitation data, the values remain identical across configurations. A revised version of Figure 5 is provided below.

475

Figure 5. Boxplots of annual values for key hydrological variables predicted by WaSiM for the 34 catchments and three configurations for the validation period (1990-1999).

• The current explanation of Figure 6 was neither informative nor relevant to my perspective. It needs to emphasise the significance of the figure.

This is a valid concern. While we recognize that it may not be of interest to all readers, we believe it is a valuable addition, as it directly links model parameters to hydrological processes, offering essential insights for WaSiM users. In order to maintain a concise article, we moved Figure 6 to the Appendix D. This approach allows us to reduce the length of the main text and ensure that readers who are particularly interested in these modeling details can still access the figure. The description text of Figure 6 was removed.

- o Figure 7, is the x-axis long-term mean of Q, or are they for a given year? The problem is between October to December in validation period. In the rest, I see no significant differences. Maybe you could drop this figure.
- 495 Figure 7 presents the mean annual hydrographs for the calibration and validation periods. As noted, all three configurations show discrepancies with observed streamflow between October and December during the validation period. Given that the figure does not provide significant additional insights and the manuscript is already lengthy, it was removed to streamline the article.

Also, since you have gaps in some of the frozen months (L130), how did you consider them in the likely monthly discharge time series?

This is a good point. Not all catchments had missing data during the frozen months. To minimize the impact of missing data, we selected calibration and validation years to ensure most catchments had complete records. With this approach, only three out of the 34 catchments had missing data. For these specific catchments, we adjusted the calibration and validation periods to focus on years without gaps. As a result, the final analysis does not include years with missing data. A sentence was added to clarify how the gaps in observed streamflow was considered at L277. Here is the added text:

"To minimize the impact of missing streamflow data, calibration and validation years were selected to ensure that most catchments had complete records."

• Language and Style:

490

500

505

510

515

The manuscript is generally well-written, but there are occasional instances of complex sentence structures that could be simplified for clarity. Thank you for acknowledging the use of ChatGPT-4. The presence of long sentences with numerous commas can be indicative of revised text by an LLM-AI.

This is a fair point. A thorough verification was conducted to ensure the text remains clear and concise, and the paper was revised by native English speakers to ensure the syntax is less "LLM-y"

- Conclusion:
  - o I remain uncertain about the meaning of lines 659-660.

Line 659 was revised to enhance clarity. The original sentence: "leads to a more accurate representation of hydrological variables."

Was modified to: "leads to a representation of hydrological processes that better aligns with expected system behavior."

This study contributes to hydrological modelling by demonstrating the importance of incorporating internal state variables, particularly groundwater recharge, into model calibration. The authors designed their model well and developed it to have three configurations and further calibrations.

The findings highlight the potential for improved representation of hydrological processes, which is crucial for water resource management and climate adaptation strategies. However, addressing the major and minor comments outlined above would further strengthen the manuscript and enhance its impact on the scientific community. Overall, with appropriate revisions, this paper has the potential to be an important addition to the literature on hydrological model calibration and process representation.

We thank the reviewer once again for the constructive review, it is much appreciated.

Sincerely,

Frédéric Talbot, on behalf of all authors

530

535

540

545

550

525

**CC1 (https://doi.org/10.5194/egusphere-2024-3353-CC1):**

The integration of internal hydrological state variables, particularly groundwater recharge, into model calibration is commendable and addresses key limitations in traditional hydrological modeling.

Thank you for the insightful comments and suggestions on our manuscript. We greatly appreciate the time and effort invested in reviewing our work. Below, we provide detailed responses to each of the comments and outline the corresponding revisions.

The chosen weights for the constrained Kling-Gupta efficiency (e.g., 70% KGE, 20% recharge standard deviation) appear somewhat arbitrary. A sensitivity analysis to justify these weights would enhance the study's robustness.

We acknowledge that the manuscript would benefit from a more detailed explanation of how the weights for the constrained Kling-Gupta efficiency were chosen. These values were determined through a trial-and-error approach, with the primary objective of integrating recharge in a realistic way to develop and demonstrate this novel methodology. Specifically, assigning 20% to the standard deviation of recharge and 10% to the mean recharge provided a balanced trade-off, ensuring recharge values remained realistic while maintaining acceptable KGE scores. For calibration, a weight of 4% on the recharge standard deviation was sufficient to preserve adequate recharge estimates while achieving strong KGE values.

While it is true that alternative hyperparameter values, as well as other objective functions, could have been tested, this would not have changed the core purpose of the study: to demonstrate that including a constraint on a physical process (in this case, recharge) can improve the representation of other processes. A comprehensive sensitivity analysis of these weights or objective functions is beyond the scope of this study but could be explored further in future research. An explanation of the assigned weight in the objective function was provided at L349-354 of section 2.5.2. Here is the original text:

"This specific weighting was chosen based on preliminary tests, where various weight combinations were evaluated on a test catchment."

Here is the new version:

"This specific weighting was determined based on preliminary testing conducted on two test catchments, where various weight combinations were evaluated. The selected weights provided the best trade-off, ensuring that recharge estimates remained realistic while maintaining strong KGE values for streamflow. In particular, assigning 20% to the recharge standard deviation and 10% to the mean annual recharge allowed the model to better capture recharge variability without compromising overall streamflow performance."

555

While the paper briefly mentions equifinality, a more in-depth exploration of how incorporating internal state variables addresses this challenge would strengthen the theoretical contribution.

The study is fundamentally centered on addressing equifinality by incorporating internal state variables. Our study design, which includes the GW and GW-RC configurations, is made to assess the effects on equifinality of introducing recharge in addition to streamflow in model calibration.

As stated in response to RC1, both GW and GW-RC configurations operate under the same model complexity, with GW calibrated on streamflow only and GW-RC utilizing both streamflow and recharge during calibration. This setup is intended to explicitly isolate and compare the impact of including recharge alongside streamflow in the calibration process.

To improve the distinction between configurations GW and GW-RC, this sentence was added at the beginning of section 2.4.3 at L311.

"Importantly, GW-RC uses the same model structure as GW, with the goal of isolating the effect of adding groundwater recharge in calibration."

The high computational demands of the GW-RC configuration are not discussed in detail. Including a section on computational trade-offs would provide valuable insights for practitioners.

We recognize the importance of discussing the computational trade-offs associated with each configuration of the WaSiM model. As stated in response to RC2, to address this, we have enhanced Table 5 to include the computational demand for each configuration, expressed in CPU-years. This addition provides a clearer understanding of the resource requirements and allows practitioners to assess the trade-offs between model complexity and computational cost.

- Data assimilation is a powerful technique widely used to integrate observations into hydrological models, improving predictions by dynamically updating model states. In this study, the authors propose an innovative calibration approach focusing on internal state variables, which aligns well with the goals of improved process representation. However, the absence of a discussion or application of data assimilation leaves an unexplored opportunity to further enhance the model's performance, then I strongly suggest to cite below papers:
- "assimilation of Sentinel-based leaf area index for surface-groundwater interaction modeling in irrigation districts"

  'Multivariate Assimilation of Satellite-based Leaf Area Index and Ground-based River Streamflow for Hydrological Modeling of Irrigated Watersheds using SWAT+'

This is an interesting suggestion. However, data assimilation is typically used to dynamically update model states based on real-time observations, which is not directly applicable in our study since we are developing a model for climate change impact

assessment where no future observations exist. Our approach focuses on improving process representation through calibration using internal state variables, ensuring that the model remains physically consistent under different climatic conditions.

Nonetheless, we recognize the relevance of data assimilation in other hydrological modeling contexts.

Thank you once again for your constructive review.

Sincerely,

590 Frédéric Talbot on behalf of all authors

---

## Author Response (AR2)

**Enhancing physically based and distributed hydrological model calibration through internal state variable constraints**

Frédéric Talbot1, Jean-Daniel Sylvain2, Guillaume Drolet2, Annie Poulin1, Richard Arsenault1

- 1 Hydrology, Climate and Climate Change Laboratory, École de technologie supérieure, Université du Québec, Montréal, H3C 1K3, Canada
- 2 Direction de la recherche forestière, Ministère des Ressources naturelles et des Forêts, Québec, G1P 3W8, Canada *Correspondence to*: Frédéric Talbot (frederic talbot.2@ens.etsmtl.ca)

**Editor's comments**

Dear Authors

Three reviewers reviewed the adjusted manuscript and while the overall manuscript improvement improved there are some issues to be addressed. Two reviewers raised concern regarding the calibration / calibration strategy. Please clarify these points and/or discuss shortcomings of the approach in the discussion as I understand the point reviewer 2 raised.

Sincerely,

Albrecht Weerts

- 15 We thank the editor for summarizing the remaining concerns and for emphasizing the need for a fair comparison between GW and GW-RC. Both configurations share the same model structure, forcing, calibration parameters, number of evaluations, and algorithm. The only differences are the inclusion of the recharge term in GW-RC's objective function and the two-step calibration required to incorporate recharge constraints in the absence of high-resolution recharge observations.
- We now explicitly acknowledge in Section 4.4 that this two-step procedure introduces a minor asymmetry, which may limit the extent to which the comparison is a fully controlled experiment. We also note that with improved recharge datasets, future work could implement a single-step calibration for both configurations.

These clarifications, along with the expanded discussion in Section 4.2 and the inclusion of the final calibrated parameter sets in Appendix D, directly address the methodological and transparency concerns raised by the reviewers.

Thank you for your review.

25 Sincerely,

Frédéric Talbot on behalf of all authors

**Anonymous referee #1**

I appreciate the clarification that the GW and GW-RC configurations share the same model complexity and differ only in calibration targets. However, my core concern remains: to rigorously isolate the effect of adding an internal state variable constraint (groundwater recharge), both configurations must be calibrated under truly identical condition, same model, data, and algorithm, except objective function for the presence or absence of the recharge term. Without this "apples-to-apples"

setup, observed differences may reflect changes in calibration design rather than the actual benefit of the recharge constraint. Even small differences in objective functions or procedures can confound interpretation.

To assess causality in calibration and equifinality studies, it is essential to vary only one factor at a time. Hydrological modeling literature emphasizes designing controlled comparisons where only one factor (e.g. model structure or calibration strategy) is varied at a time (Clark et al., 2015). For example, Pool et al. (2025) calibrated the same model with discharge only, evapotranspiration only, and both together, explicitly isolating the effect of multi-variable calibration under identical setups. Similarly, calibrating GW and GW-RC with the same objective structure (except for the recharge term) would allow direct assessment of how internal constraints influence parameter identifiability and simulation realism.

We thank the reviewer for this comment and for highlighting the importance of ensuring a controlled comparison between configurations. In our study, configurations GW and GW-RC were explicitly designed to share the same model structure and complexity, with identical calibration parameters, number of evaluations, and calibration algorithm. The sole differences are in the objective function, where GW-RC includes groundwater recharge alongside streamflow, and in the parameter space during calibration. While it is not entirely clear which specific aspect of the methodology the reviewer is concerned about, we suspect it may relate to the two-step calibration approach applied in the GW-RC configuration.

For GW-RC, a two-step calibration was required to incorporate groundwater recharge. In the first step, regional recharge estimates derived from available large-scale data were used to constrain the range of five recharge-sensitive parameters. In the second step, the model was recalibrated using streamflow and the recharge standard deviation, allowing each catchment to freely adjust its recharge rates within the constrained parameter space. This approach allows the model to adapt recharge estimates to each catchment's specific conditions, preventing the regional estimates from exerting disproportionate influence on the final recharge outcomes. This two-step process is the only practical way to integrate recharge constraints in our study area without imposing unrealistic values. Conversely, configuration GW could not be calibrated with this two-step procedure, as it does not use recharge constraints.

We acknowledge that the optimal "apples-to-apples" setup described by the reviewer, where only the objective function differs, would require high-quality, high-resolution recharge observations to allow a single-step calibration. Such data are not available for our study region, and this limitation is common in many other regions where this type of methodology could be applied. In this context, the two-step procedure we propose represents a practical solution to address the absence of complete recharge observations. Regional recharge estimates can serve as an a priori constraint, which the model then optimizes based on streamflow data (and groundwater recharge standard deviation) at the catchment scale. Future work benefiting from improved recharge datasets could implement a single-step calibration using both streamflow and recharge, enabling a more direct comparison between configurations. To address this, we have added a few sentences in Section 4.4 explicitly acknowledging the limitation of our two-step approach and noting that future work with improved recharge observations could adopt a one-step calibration for a more direct comparison between configurations.

Here are the added sentences:

65 "The two-step calibration adopted for GW-RC, necessitated by the absence of high-quality, spatially distributed recharge observations, limits the extent to which a fully direct comparison with GW can be achieved. Future work with access to such datasets could implement a single-step calibration using both streamflow and recharge, enabling a more controlled assessment of the effects of internal recharge constraints."

Equifinality is central here: calibrating to streamflow alone often yields many parameter sets that produce similar outputs but divergent internal processes (Pool et al., 2025). Including internal data like recharge helps reduce equifinality by narrowing the feasible parameter space. Gallart et al. (2007) demonstrated this clearly showing that internal catchment observations reduced uncertainty in discharge and baseflow predictions. But for such added value to be credibly demonstrated, the comparison must be fair. Without symmetric calibration design (e.g., calibrating GW-RC using streamflow only as well), conclusions about the "effectiveness" of recharge constraints remain suggestive, not definitive.

We thank the reviewer for emphasizing the importance of equifinality in hydrological modeling and for noting that incorporating internal state variables like recharge can help reduce parameter uncertainty. In our study, configuration GW is identical to configuration GW-RC except for the inclusion of the recharge term in the objective function and the associated two-step procedure used to define the parameter space. Both configurations share the same model structure, complexity, calibration parameters, number of evaluations, and optimization algorithm. The only differences are those directly related to integrating groundwater recharge, which we have described in detail in our methodology.

The very purpose of GW-RC is to test the added value of including recharge information in calibration. Removing recharge from the objective function would negate the defining feature of this configuration and transform it into configuration GW. Instead, our approach uses GW as the "streamflow only" baseline and GW-RC as the "streamflow plus recharge" configuration, ensuring that the difference between them reflects the influence of recharge constraints.

We also recognize that the use of a two-step calibration in GW-RC, necessitated by the lack of high-resolution spatially distributed recharge observations, introduces a minor asymmetry in the calibration design. As noted in our revised manuscript, future work could address this by using high-quality recharge observations in a single-step calibration for both configurations, allowing an even more direct and controlled assessment of how recharge constraints influence equifinality and simulation realism.

In short, my call for a more controlled experiment is not about simplifying the setup, but enabling causal inference. Holding model structure and forcing constant while toggling the recharge constraint is the only way to quantify its true benefit or trade-off. Multi-objective calibration is praised for reducing equifinality, but its effectiveness must be benchmarked against an identical single-objective case. I strongly encourage the authors to consider recalibrating GW and GW-RC on commensurate terms to ensure that the observed improvements in GW-RC are truly attributable to internal constraints, and not to differences in calibration setup.

**References**

Clark, M. P., et al. (2015). A unified approach for process-based hydrologic modeling: 1. Modeling concept. Water Resources Research, 51(4), 2498–2514.

Pool, S., Fowler, K., Gardiya Weligamage, H., & Peel, M. (2025). Multivariate calibration can increase simulated discharge uncertainty and model equifinality. EGUsphere. https://doi.org/10.5194/egusphere-2025-1598

Gallart, F., Latron, J., Llorens, P., & Beven, K. (2007). Using internal catchment information to reduce the uncertainty of discharge and baseflow predictions. Advances in Water Resources, 30(4), 808–823.

We thank the reviewer for this valuable suggestion and for emphasizing the importance of causal inference in evaluating the benefits of recharge constraints. In our study, GW and GW-RC are identical in model structure, forcing, calibration parameters, number of evaluations, and optimization algorithm, with differences arising solely from the inclusion of the recharge term in the objective function and the two-step calibration process required to integrate it. We acknowledge that the most direct way to achieve a fully symmetric calibration design would be to recalibrate GW-RC using streamflow only. However, this would remove the defining characteristic of GW-RC and effectively reproduce the GW configuration. As discussed in our revised manuscript, the asymmetry introduced by the two-step calibration is an inherent consequence of working without high-110 resolution, spatially distributed recharge observations. Future work with such data could apply a single-step calibration to both configurations, enabling the perfectly controlled comparison the reviewer describes.

Thank you once again for your constructive review.

Sincerely,

Frédéric Talbot on behalf of all authors

**115 Referee #2**

100

105

Thank you for the thorough revisions and thoughtful responses to my comments. I appreciate the considerable effort the authors have put into improving the manuscript. The changes, particularly the enhanced abstract, more precise explanation of model configurations, expanded methodological details, and enhanced discussion on model performance and calibration strategy, have significantly strengthened the clarity and scientific value of the work.

120 I wish the authors all the best in their future scientific endeavours.

I would like to make one final optional minor suggestion regarding Figure 5. Since precipitation is the same across all model setups, it may be visually more apparent to display only a single box per catchment for precipitation—perhaps using a neutral colour like black—rather than repeating it for each configuration. This adjustment would reduce visual redundancy and help emphasise the differences between model outputs, and water balance closure.

125 Thank you again for addressing the comments so carefully.

Sincerely,

130

We thank the reviewer for the thoughtful and encouraging feedback, as well as for the valuable suggestion regarding Figure 5. We agree that displaying a single precipitation box per catchment using a neutral colour improves the visual clarity of the figure and better highlights the differences among model outputs and water balance closure. Figure 5 has been revised accordingly. Here is the new Figure 5:

Thank you once again for your constructive review and supportive comments.

Sincerely,

Frédéric Talbot on behalf of all authors

**Anonymous referee #3**

140

145

150

155

160

165

The manuscript addresses a significant issue in hydrological modeling, specifically focusing on the calibration of distributed, physically based models by integrating groundwater recharge constraints. The authors demonstrate how incorporating additional calibration constraints beyond streamflow can enhance internal process representations despite minor trade-offs in conventional performance metrics. This study presents valuable insights into the implications of equifinality in hydrological modeling and the benefits of multi-objective calibration approaches.

The authors have adequately addressed the comments raised by the previous three reviewers, resulting in a manuscript with a strong organization and clear presentation. However, I identify one major issue that requires further clarification from the authors: In Section 2.3.2, the authors state that a total of 17 parameters were calibrated in this study. It remains unclear whether all 17 parameters were re-calibrated independently for each configuration (BL, GW, GW-RC), or if some parameters were held constant while only a subset was re-calibrated.

We thank the reviewer for the thoughtful and encouraging comments, as well as for highlighting the need for clarification regarding the calibration process. All 17 parameters were recalibrated independently for each configuration (BL, GW, GW-RC), with no parameters held constant between configurations. We agree that this point could be made clearer in the manuscript and have added the following sentence to Section 2.3.2:

"For each model configuration (BL, GW, GW-RC), the full set of 17 parameters was recalibrated independently within the specified ranges."

If all 17 parameters were recalibrated independently for each configuration, the authors should further discuss the implications of incorporating groundwater recharge constraints on the entire parameter set. Specifically, do the groundwater recharge constraints influence parameter values even in seemingly unrelated sub-models, such as those controlling snowmelt and evapotranspiration processes?

If only a subset of parameters were re-calibrated and the remaining parameters were kept fixed across configurations, the authors should explicitly specify which parameters were held constant and clearly justify their rationale for this choice.

We thank the reviewer for the insightful question regarding the broader influence of incorporating groundwater recharge constraints on the full parameter set. All 17 calibration parameters were recalibrated independently for each configuration. While Section 4.2 already discusses the effects of recharge constraints on key parameters such as *QDsnow* and drainage density, we agree that the implications for other parameters warrant further discussion. To address this, we have added the following paragraph at the end of Section 4.2:

"Moreover, configuration GW-RC also exhibited lower values of  $k_h$  (storage coefficient for interflow), higher values of Krec (recession constant for hydraulic conductivity), lower correction factors for PET in summer, and higher correction factors for PET in winter compared to the other two configurations. These differences indicate that adding groundwater recharge constraints during calibration can influence parameter values in sub-models that are seemingly unrelated to groundwater processes, such as evapotranspiration. This suggests that the recharge constraint propagates through the model structure,

affecting multiple hydrological components. A complete list of calibrated parameter values for each catchment and configuration is provided in Appendix D."

Additionally, I suggest that the authors include a supplementary document listing the final calibrated parameter sets for all catchments across each configuration. This would allow readers to intuitively compare parameter differences between configurations, facilitating greater understanding and reproducibility of the results.

This is an excellent suggestion. In response, we have included a new table in Appendix D presenting the final calibrated parameter values for all catchments across each configuration (BL, GW, GW-RC). This addition enables readers to directly compare parameter values between configurations.

170

| Catchment |                            | k D k H |      |      |      |      | d r |    |    | QD Snow |     |     | c 0 |     |     | K rec |     |     | T 0 |      |      | T R/S |      |      | C WH |     |     |   |
|-----------|----------------------------|-------------------------------|------|------|------|------|----------------|----|----|--------------------|-----|-----|----------------|-----|-----|------------------|-----|-----|----------------|------|------|------------------|------|------|-----------------|-----|-----|---|
| Code      | Name                       | Α                             | В    | С    | Α    | В    | С              | Α  | В  | С                  | Α   | В   | С              | Α   | В   | С                | Α   | В   | С              | Α    | В    | С                | Α    | В    | С               | Α   | В   | ( |
| 1         | Bonaventure                | 24.7                          | 24.7 | 25.0 | 25.0 | 24.8 | 24.5           | 38 | 46 | 19                 | 0.1 | 0.1 | 0.4            | 1.8 | 2.2 | 2.6              | 0.5 | 0.1 | 0.9            | -0.6 | -0.1 | 0.4              | 0.3  | 0.5  | 0.0             | 0.1 | 0.1 | 0 |
| 2         | York                       | 25.0                          | 24.8 | 24.7 | 24.1 | 24.3 | 14.9           | 38 | 18 | 5                  | 0.1 | 0.1 | 8.0            | 1.2 | 1.8 | 1.4              | 0.6 | 0.4 | 0.3            | -1.3 | -0.2 | 0.3              | 1.3  | 0.2  | 1.6             | 0.1 | 0.1 | 0 |
| 3         | Dartmouth                  | 21.6                          | 24.4 | 24.7 | 20.9 | 17.7 | 1.8            | 49 | 50 | 8                  | 0.1 | 0.1 | 8.0            | 2.1 | 2.1 | 2.1              | 0.8 | 0.9 | 0.2            | -0.3 | -0.1 | 0.1              | 1.4  | 1.0  | 2.3             | 0.1 | 0.1 | ( |
| 4         | Matane                     | 24.7                          | 24.9 | 24.6 | 25.0 | 24.7 | 19.8           | 44 | 48 | 20                 | 0.1 | 0.1 | 0.6            | 2.3 | 2.9 | 2.5              | 0.1 | 0.2 | 0.2            | -0.1 | 0.6  | 0.6              | 1.8  | 0.7  | 1.1             | 0.1 | 0.1 | ( |
| 5         | Rimouski                   | 24.8                          | 24.9 | 24.6 | 24.9 | 25.0 | 23.8           | 43 | 17 | 4                  | 0.2 | 0.3 | 0.4            | 1.5 | 1.9 | 2.1              | 0.2 | 0.2 | 0.9            | -0.1 | 0.6  | -0.3             | 0.3  | -2.4 | 2.5             | 0.2 | 0.2 | ( |
| 6         | Des Trois-Pistoles         | 24.7                          | 24.7 | 25.0 | 24.9 | 24.5 | 1.6            | 49 | 48 | 2                  | 0.2 | 0.2 | 8.0            | 1.9 | 2.8 | 2.1              | 1.0 | 1.0 | 0.5            | 0.2  | 1.0  | 0.7              | -3.1 | -2.6 | -2.3            | 0.3 | 0.2 | - |
| 7         | Du Loup                    | 24.7                          | 23.4 | 24.9 | 24.3 | 24.1 | 16.8           | 41 | 8  | 4                  | 0.1 | 0.2 | 0.6            | 2.0 | 2.6 | 1.9              | 0.2 | 0.2 | 0.9            | -0.8 | 0.7  | 0.6              | 1.7  | 0.3  | -2.6            | 0.2 | 0.3 | - |
| 8         | Ouelle                     | 24.7                          | 24.9 | 24.6 | 24.6 | 23.6 | 3.5            | 40 | 50 | 19                 | 0.1 | 0.1 | 0.6            | 2.3 | 2.5 | 2.3              | 0.1 | 0.1 | 1.0            | -0.1 | 0.0  | 0.6              | 1.7  | 1.8  | 1.2             | 0.3 | 0.3 | - |
| 9         | Famine                     | 22.6                          | 24.5 | 25.0 | 19.8 | 11.8 | 1.6            | 46 | 48 | 18                 | 0.1 | 0.1 | 0.7            | 1.9 | 2.7 | 2.6              | 1.0 | 0.7 | 0.3            | -1.1 | -0.8 | 0.7              | 1.1  | 1.2  | 0.3             | 0.2 | 0.3 |   |
| 10        | Bécancour                  | 24.9                          | 24.4 | 24.7 | 23.6 | 25.0 | 23.5           | 35 | 47 | 20                 | 0.1 | 0.1 | 0.7            | 2.7 | 2.5 | 2.0              | 0.1 | 0.9 | 0.9            | 0.2  | 0.2  | 0.0              | 1.3  | 1.3  | 1.7             | 0.1 | 0.1 |   |
| 11        | Nicolet Sud-Ouest          | 25.0                          | 24.1 | 24.9 | 24.9 | 24.7 | 16.4           | 49 | 36 | 33                 | 0.1 | 0.1 | 0.7            | 2.9 | 2.8 | 2.5              | 0.6 | 1.0 | 0.9            | 0.7  | 0.5  | 8.0              | -3.0 | -0.2 | -2.9            | 0.1 | 0.2 |   |
| 12        | Nicolet                    | 24.9                          | 24.7 | 24.8 | 22.8 | 19.1 | 5.0            | 25 | 45 | 28                 | 0.1 | 0.2 | 1.0            | 3.0 | 2.0 | 1.7              | 0.9 | 0.9 | 1.0            | 0.7  | -0.6 | 0.3              | -2.9 | 0.2  | 0.0             | 0.1 | 0.2 |   |
| 13        | Eaton                      | 24.7                          | 24.3 | 20.9 | 3.5  | 2.9  | 2.0            | 20 | 49 | 9                  | 0.3 | 0.4 | 1.0            | 2.8 | 3.0 | 1.8              | 0.9 | 0.9 | 0.2            | 0.0  | -0.1 | -0.2             | -3.6 | -0.3 | -3.8            | 0.1 | 0.2 |   |
| 14        | Au Saumon                  | 25.0                          | 24.7 | 24.9 | 20.6 | 12.7 | 2.7            | 49 | 49 | 18                 | 0.1 | 0.3 | 1.0            | 2.5 | 3.0 | 2.1              | 0.9 | 1.0 | 0.2            | -0.8 | 0.0  | 0.1              | 1.7  | -2.6 | -1.5            | 0.1 | 0.1 |   |
| 15        | Noire                      | 25.0                          | 24.5 | 24.9 | 13.8 | 14.9 | 2.5            | 27 | 45 | 21                 | 0.1 | 0.3 | 1.0            | 2.1 | 3.0 | 2.2              | 0.8 | 8.0 | 1.0            | -1.0 | 0.3  | 0.3              | 0.5  | -1.2 | 0.4             | 0.3 | 0.3 |   |
| 16        | Rouge                      | 17.8                          | 23.5 | 24.3 | 24.8 | 24.7 | 24.8           | 40 | 5  | 6                  | 0.1 | 0.1 | 0.3            | 0.8 | 1.7 | 0.9              | 0.1 | 0.9 | 0.2            | -2.3 | -0.8 | -3.2             | 0.1  | 0.7  | 1.9             | 0.1 | 0.2 |   |
| 17        | Gatineau                   | 24.1                          | 22.6 | 24.9 | 12.7 | 24.3 | 24.8           | 19 | 5  | 6                  | 0.3 | 0.3 | 0.6            | 0.9 | 1.7 | 0.9              | 0.3 | 0.2 | 0.7            | -3.0 | -0.1 | 0.0              | 3.1  | -3.8 | -1.9            | 0.3 | 0.3 |   |
| 18        | Kinojévis                  | 24.9                          | 22.9 | 24.0 | 24.4 | 24.2 | 21.4           | 46 | 19 | 7                  | 0.5 | 0.1 | 0.5            | 1.1 | 1.8 | 0.9              | 0.8 | 0.3 | 0.8            | 0.9  | 0.5  | -1.9             | -3.2 | -0.9 | 3.5             | 0.3 | 0.1 |   |
| 19        | Mattawin                   | 22.6                          | 24.1 | 24.7 | 24.8 | 24.9 | 20.9           | 25 | 15 | 4                  | 0.1 | 0.1 | 0.6            | 1.7 | 2.2 | 0.8              | 0.1 | 0.1 | 0.6            | -0.3 | 1.0  | -1.3             | 0.8  | -1.7 | 1.3             | 0.2 | 0.1 |   |
| 20        | Croche                     | 9.3                           | 11.1 | 24.8 | 21.6 | 13.7 | 24.8           | 25 | 1  | 17                 | 0.1 | 0.1 | 0.4            | 0.8 | 2.2 | 1.0              | 0.3 | 0.3 | 0.2            | -1.2 | 1.3  | -0.2             | -0.5 | -2.2 | -3.2            | 0.3 | 0.2 |   |
| 21        | Vermillon                  | 12.7                          | 12.2 | 24.7 | 20.1 | 21.5 | 23.9           | 12 | 5  | 1                  | 0.2 | 0.1 | 0.5            | 1.0 | 1.9 | 1.2              | 0.3 | 0.3 | 0.3            | -0.5 | 0.5  | 0.5              | -3.2 | 0.9  | 0.7             | 0.3 | 0.2 |   |
| 22        | Batiscan                   | 22.6                          | 22.5 | 24.9 | 24.5 | 24.8 | 1.8            | 18 | 25 | 1                  | 0.2 | 0.1 | 0.8            | 0.7 | 0.5 | 0.7              | 0.2 | 0.2 | 0.6            | -1.9 | -2.1 | 0.2              | -1.8 | -3.0 | -1.7            | 0.2 | 0.3 |   |
| 23        | Sainte-Anne                | 24.9                          | 24.2 | 24.9 | 23.6 | 18.5 | 7.3            | 34 | 25 | 23                 | 0.3 | 0.3 | 0.8            | 1.7 | 1.4 | 1.0              | 0.2 | 0.3 | 0.9            | 0.3  | 0.0  | -0.1             | -3.2 | -0.2 | -0.4            | 0.1 | 0.1 |   |
| 24        | Bras du Nord               | 24.7                          | 24.9 | 25.0 | 24.4 | 24.3 | 24.2           | 34 | 46 | 22                 | 0.1 | 0.1 | 0.4            | 1.4 | 1.0 | 1.3              | 0.7 | 0.2 | 0.7            | 0.4  | -1.5 | 0.3              | -3.4 | -1.2 | -0.7            | 0.1 | 0.2 |   |
| 25        | Ouareau                    | 21.8                          | 23.0 | 24.7 | 24.8 | 16.3 | 24.7           | 39 | 7  | 12                 | 0.2 | 0.1 | 0.5            | 1.6 | 2.5 | 1.6              | 0.1 | 0.1 | 0.6            | 0.5  | 0.8  | 0.7              | -1.2 | 0.5  | -1.9            | 0.2 | 0.2 |   |
| 26        | L'Assomption               | 24.7                          | 22.5 | 24.8 | 24.3 | 24.5 | 25.0           | 33 | 31 | 13                 | 0.1 | 0.1 | 0.4            | 1.1 | 2.3 | 1.6              | 0.1 | 0.1 | 0.7            | -0.8 | 0.3  | 0.7              | 0.8  | -1.0 | -0.2            | 0.1 | 0.1 |   |
| 27        | De l'Achigan               | 24.9                          | 23.6 | 24.6 | 24.7 | 1.6  | 6.0            | 44 | 48 | 11                 | 0.1 | 0.1 | 0.4            | 2.2 | 2.8 | 2.7              | 0.2 | 0.6 | 0.8            | 0.3  | 0.6  | 0.6              | -2.8 | -0.2 | -2.1            | 0.1 | 0.2 |   |
| 28        | Du Loup                    | 24.9                          | 23.4 | 24.5 | 24.9 | 24.7 | 24.3           | 19 | 19 | 3                  | 0.2 | 0.1 | 0.4            | 1.0 | 1.3 | 1.6              | 0.4 | 0.2 | 0.1            | -0.8 | 0.6  | 0.3              | 0.0  | -0.4 | 2.8             | 0.2 | 0.2 |   |
| 29        | Petit Saguenay             | 24.9                          | 24.6 | 24.8 | 24.8 | 3.6  | 2.7            | 22 | 9  | 2                  | 0.1 | 0.2 | 0.7            | 1.5 | 1.8 | 1.0              | 0.1 | 0.3 | 0.9            | 0.2  | 0.0  | -0.1             | 0.3  | -3.8 | 1.3             | 0.1 | 0.2 |   |
| 30        | Petite rivière Péribonca   | 24.7                          | 24.3 | 24.8 | 23.6 | 24.4 | 1.2            | 40 | 26 | 4                  | 0.1 | 0.1 | 0.8            | 1.7 | 0.7 | 0.5              | 0.3 | 0.3 | 0.7            | 1.5  | -3.5 | -0.8             | -3.1 | 2.4  | 0.2             | 0.1 | 0.3 |   |
| 31        | Métabetchouane             | 24.2                          | 24.8 | 25.0 | 24.4 | 18.0 | 16.5           | 26 | 9  | 1                  | 0.1 | 0.2 | 0.5            | 0.6 | 1.4 | 1.1              | 0.1 | 0.1 | 0.5            | -2.2 | -1.1 | 1.3              | -0.4 | -0.8 | -2.5            | 0.3 | 0.3 |   |
| 32        | Valin                      | 25.0                          | 22.8 | 25.0 | 23.5 | 19.4 | 17.2           | 44 | 10 | 7                  | 0.2 | 0.1 | 0.9            | 1.2 | 2.4 | 0.8              | 0.1 | 0.5 | 0.9            | 0.7  | 1.4  | 0.4              | -2.5 | 1.2  | 0.3             | 0.1 | 0.1 |   |
| 33        | Sainte-Marguerite Nord-Est | 25.0                          | 24.7 | 24.6 | 24.3 | 17.5 | 18.2           | 39 | 11 | 12                 | 0.1 | 0.3 | 0.9            | 1.7 | 2.4 | 1.3              | 0.1 | 0.3 | 0.9            | 0.6  | 1.1  | 0.7              | -2.6 | -3.5 | -0.9            | 0.1 | 0.1 |   |
| 34        | Godbout                    | 24.8                          | 23.2 | 24.8 | 23.8 | 24.7 | 24.6           | 44 | 21 | 20                 | 0.3 | 0.1 | 0.9            | 1.3 | 2.4 | 1.4              | 1.0 | 0.3 | 0.6            | 0.1  | 0.6  | 0.5              | 0.7  | 1.0  | -1.8            | 0.2 | 0.2 |   |
|           | Average                    | 23.4                          | 23.3 | 24.6 | 22.6 | 19.9 | 14.6           | 35 | 28 | 12                 | 0.2 | 0.2 | 0.7            | 1.7 | 2.1 | 1.6              | 0.4 | 0.4 | 0.6            | -0.4 | 0.0  | 0.1              | -0.6 | -0.5 | -0.2            | 0.2 | 0.2 |   |

|      | C rfr           |     |     | f i,summer |     |     | f i,fall |     |     | f i,winter |     |     | f i,spring |     |     | Kol |   |    |    | K XY |     | d z |     |     |     |
|------|----------------------------|-----|-----|-----------------------|-----|-----|---------------------|-----|-----|-----------------------|-----|-----|-----------------------|-----|-----|-----|---|----|----|-----------------|-----|----------------|-----|-----|-----|
| Code | Name                       | Α   | В   | С                     | Α   | В   | С                   | Α   | В   | С                     | Α   | В   | С                     | Α   | В   | С   | Α | В  | С  | Α               | В   | С              | Α   | В   | С   |
| 1    | Bonaventure                | 0.7 | 0.2 | 0.7                   | 1.3 | 1.1 | 1.0                 | 0.5 | 0.2 | 0.6                   | 0.5 | 2.0 | 1.5                   | 1.4 | 0.8 | 0.7 | - | 83 | 9  | -               | 3.2 | 1.4            | 1.1 | 1.3 | 1.4 |
| 2    | York                       | 0.6 | 0.6 | 8.0                   | 1.7 | 1.1 | 0.5                 | 1.0 | 0.6 | 0.2                   | 0.2 | 0.2 | 1.3                   | 1.6 | 1.4 | 1.8 | - | 5  | 15 | -               | 0.6 | 1.5            | 1.1 | 1.3 | 1.4 |
| 3    | Dartmouth                  | 0.6 | 0.9 | 8.0                   | 1.2 | 1.2 | 0.9                 | 1.4 | 0.7 | 0.7                   | 0.3 | 0.2 | 0.9                   | 0.6 | 0.7 | 0.6 | - | 35 | 55 | -               | 0.2 | 1.7            | 0.9 | 1.0 | 1.3 |
| 4    | Matane                     | 0.6 | 0.6 | 0.9                   | 1.2 | 1.6 | 1.3                 | 1.7 | 8.0 | 0.9                   | 0.4 | 1.0 | 1.0                   | 0.9 | 0.2 | 0.5 | - | 97 | 44 | -               | 3.2 | 1.6            | 1.1 | 1.1 | 0.9 |
| 5    | Rimouski                   | 0.6 | 0.6 | 0.7                   | 1.2 | 1.2 | 0.9                 | 1.7 | 8.0 | 0.7                   | 0.3 | 8.0 | 0.2                   | 0.9 | 0.7 | 1.1 | - | 23 | 35 | -               | 0.3 | 0.5            | 1.1 | 1.1 | 8.0 |
| 6    | Des Trois-Pistoles         | 0.7 | 0.4 | 0.4                   | 1.3 | 1.3 | 1.2                 | 1.2 | 1.5 | 1.1                   | 0.1 | 0.3 | 0.6                   | 1.0 | 0.6 | 0.7 | - | 44 | 56 | -               | 1.5 | 2.4            | 1.4 | 8.0 | 8.0 |
| 7    | Du Loup                    | 0.7 | 1.0 | 0.7                   | 1.8 | 1.2 | 1.1                 | 1.4 | 0.6 | 8.0                   | 0.4 | 0.1 | 8.0                   | 8.0 | 1.1 | 1.0 | - | 20 | 41 | -               | 0.2 | 2.3            | 1.1 | 1.3 | 1.2 |
| 8    | Ouelle                     | 0.7 | 8.0 | 1.0                   | 1.1 | 1.1 | 1.1                 | 2.0 | 1.5 | 1.0                   | 0.7 | 1.0 | 1.2                   | 1.1 | 0.9 | 0.9 | - | 40 | 92 | -               | 0.2 | 2.2            | 1.1 | 1.0 | 1.2 |
| 9    | Famine                     | 0.6 | 0.7 | 0.7                   | 1.0 | 1.4 | 1.0                 | 1.7 | 0.1 | 8.0                   | 0.2 | 0.5 | 0.9                   | 1.1 | 0.7 | 0.7 | - | 21 | 63 | -               | 0.2 | 0.6            | 1.2 | 0.9 | 1.1 |
| 10   | Bécancour                  | 0.6 | 1.0 | 0.6                   | 1.5 | 1.3 | 1.2                 | 0.6 | 1.5 | 0.7                   | 0.7 | 0.7 | 1.4                   | 0.9 | 0.7 | 0.5 | - | 47 | 75 | -               | 1.4 | 3.6            | 1.1 | 1.4 | 1.3 |
| 11   | Nicolet Sud-Ouest          | 0.7 | 0.3 | 0.9                   | 1.2 | 1.4 | 1.2                 | 1.0 | 0.5 | 0.9                   | 0.8 | 0.6 | 1.1                   | 1.1 | 0.9 | 0.9 | - | 62 | 97 | -               | 0.2 | 0.6            | 1.1 | 1.0 | 0.9 |
| 12   | Nicolet                    | 0.8 | 1.0 | 8.0                   | 1.2 | 1.7 | 1.7                 | 1.0 | 0.6 | 0.4                   | 0.9 | 0.2 | 1.3                   | 1.0 | 0.9 | 0.6 | - | 65 | 62 | -               | 0.2 | 3.9            | 1.3 | 8.0 | 1.1 |
| 13   | Eaton                      | 0.6 | 8.0 | 0.9                   | 0.9 | 1.2 | 8.0                 | 1.2 | 1.4 | 1.3                   | 0.8 | 0.4 | 1.1                   | 1.6 | 1.0 | 0.9 | - | 21 | 37 | -               | 0.2 | 2.8            | 1.2 | 8.0 | 1.2 |
| 14   | Au Saumon                  | 0.8 | 0.9 | 0.9                   | 1.0 | 1.1 | 0.9                 | 1.3 | 0.7 | 0.7                   | 0.9 | 0.1 | 0.9                   | 0.7 | 0.6 | 0.6 | - | 20 | 66 | -               | 0.2 | 3.9            | 1.3 | 8.0 | 1.2 |
| 15   | Noire                      | 0.8 | 0.7 | 0.7                   | 1.2 | 1.2 | 1.0                 | 1.1 | 1.5 | 1.6                   | 0.6 | 0.2 | 0.7                   | 1.2 | 1.0 | 0.9 | - | 64 | 94 | -               | 0.3 | 2.6            | 1.2 | 8.0 | 1.1 |
| 16   | Rouge                      | 1.0 | 0.9 | 1.0                   | 1.2 | 1.1 | 1.7                 | 0.9 | 0.9 | 0.6                   | 1.5 | 0.3 | 1.7                   | 1.0 | 0.5 | 0.2 | - | 16 | 25 | -               | 2.2 | 2.6            | 1.2 | 1.3 | 0.9 |
| 17   | Gatineau                   | 0.6 | 8.0 | 8.0                   | 1.3 | 1.2 | 8.0                 | 0.4 | 0.5 | 1.2                   | 1.9 | 0.1 | 1.3                   | 1.1 | 0.6 | 0.9 | - | 15 | 14 | -               | 2.2 | 4.0            | 1.0 | 1.3 | 1.1 |
| 18   | Kinojévis                  | 0.7 | 0.5 | 0.5                   | 0.4 | 1.0 | 0.7                 | 1.2 | 1.0 | 1.8                   | 1.0 | 0.9 | 1.3                   | 1.7 | 0.2 | 0.6 | - | 4  | 21 | -               | 3.2 | 3.9            | 1.4 | 1.1 | 1.2 |
| 19   | Mattawin                   | 0.6 | 0.7 | 0.4                   | 1.2 | 0.6 | 0.9                 | 1.4 | 1.3 | 1.7                   | 0.4 | 0.3 | 1.7                   | 1.1 | 1.0 | 8.0 | - | 14 | 21 | -               | 3.0 | 3.8            | 1.4 | 1.4 | 1.2 |
| 20   | Croche                     | 0.6 | 0.7 | 1.0                   | 1.7 | 1.2 | 1.0                 | 1.3 | 0.2 | 1.6                   | 1.4 | 0.1 | 1.3                   | 0.6 | 8.0 | 0.7 | - | 26 | 29 | -               | 8.0 | 3.6            | 0.9 | 8.0 | 1.1 |
| 21   | Vermillon                  | 0.7 | 0.5 | 0.7                   | 1.9 | 1.4 | 1.0                 | 0.5 | 1.0 | 2.0                   | 0.5 | 0.3 | 2.0                   | 1.0 | 0.9 | 0.2 | - | 4  | 21 | -               | 1.0 | 4.0            | 1.3 | 1.3 | 1.1 |
| 22   | Batiscan                   | 0.7 | 0.7 | 0.7                   | 1.6 | 1.2 | 0.4                 | 1.4 | 0.7 | 1.4                   | 0.2 | 0.5 | 1.9                   | 1.0 | 1.2 | 1.3 | - | 21 | 32 | -               | 4.0 | 3.5            | 1.1 | 1.3 | 1.0 |
| 23   | Sainte-Anne                | 0.7 | 8.0 | 0.9                   | 1.1 | 1.0 | 0.9                 | 0.6 | 0.3 | 0.6                   | 0.4 | 0.6 | 1.6                   | 1.0 | 0.8 | 0.6 | - | 7  | 40 | -               | 1.3 | 2.9            | 1.2 | 1.1 | 1.2 |
| 24   | Bras du Nord               | 0.8 | 8.0 | 8.0                   | 1.5 | 1.2 | 0.9                 | 1.0 | 0.5 | 0.5                   | 0.7 | 0.2 | 1.9                   | 1.0 | 0.9 | 0.7 | - | 3  | 33 | -               | 1.7 | 3.1            | 0.8 | 1.3 | 0.9 |
| 25   | Ouareau                    | 0.6 | 0.9 | 8.0                   | 1.4 | 1.0 | 1.0                 | 1.8 | 0.6 | 0.7                   | 1.5 | 0.5 | 1.0                   | 0.5 | 0.7 | 0.1 | - | 26 | 29 | -               | 1.0 | 3.9            | 0.9 | 1.1 | 1.2 |
| 26   | L'Assomption               | 0.8 | 8.0 | 0.3                   | 1.6 | 1.3 | 1.3                 | 0.6 | 0.7 | 1.3                   | 0.4 | 0.7 | 1.3                   | 1.3 | 0.6 | 0.5 | - | 9  | 12 | -               | 1.2 | 3.2            | 1.3 | 1.4 | 1.2 |
| 27   | De l'Achigan               | 0.7 | 0.7 | 0.7                   | 1.2 | 1.3 | 1.4                 | 0.6 | 0.4 | 0.4                   | 0.7 | 0.9 | 0.6                   | 0.9 | 0.5 | 0.4 | - | 34 | 32 | -               | 0.4 | 1.2            | 1.1 | 1.1 | 8.0 |
| 28   | Du Loup                    | 0.9 | 0.6 | 1.0                   | 1.6 | 1.2 | 1.2                 | 1.0 | 8.0 | 0.4                   | 0.4 | 0.5 | 1.5                   | 1.4 | 0.9 | 8.0 | - | 4  | 16 | -               | 4.0 | 2.6            | 1.1 | 1.2 | 1.0 |
| 29   | Petit Saguenay             | 0.7 | 0.1 | 0.3                   | 1.7 | 1.4 | 0.6                 | 1.9 | 0.9 | 1.2                   | 0.5 | 0.3 | 1.3                   | 1.3 | 0.8 | 1.5 | - | 25 | 46 | -               | 1.8 | 1.8            | 1.1 | 1.3 | 1.2 |
| 30   | Petite rivière Péribonca   | 1.0 | 1.0 | 0.4                   | 1.7 | 1.1 | 0.4                 | 1.9 | 8.0 | 1.6                   | 0.3 | 1.7 | 2.0                   | 1.9 | 0.9 | 1.5 | - | 6  | 35 | -               | 3.7 | 4.0            | 1.3 | 1.1 | 1.3 |
| 31   | Métabetchouane             | 0.7 | 0.9 | 0.7                   | 1.5 | 1.2 | 0.7                 | 1.0 | 1.2 | 1.7                   | 1.2 | 0.4 | 1.6                   | 1.2 | 0.2 | 1.2 | - | 3  | 28 | -               | 3.2 | 3.8            | 1.1 | 1.2 | 1.0 |
| 32   | Valin                      | 0.6 | 0.7 | 8.0                   | 1.4 | 0.7 | 0.1                 | 0.2 | 0.7 | 0.3                   | 0.3 | 0.2 | 0.7                   | 0.9 | 0.4 | 1.2 | - | 12 | 77 | -               | 2.2 | 3.9            | 1.1 | 1.1 | 1.2 |
| 33   | Sainte-Marguerite Nord-Est | 0.7 | 8.0 | 0.4                   | 0.9 | 0.6 | 0.1                 | 0.4 | 0.3 | 0.6                   | 0.1 | 0.2 | 1.0                   | 0.9 | 0.3 | 8.0 | - | 27 | 80 | -               | 1.3 | 2.8            | 1.1 | 1.3 | 1.2 |
| 34   | Godbout                    | 0.7 | 0.6 | 0.9                   | 1.0 | 0.7 | 0.2                 | 8.0 | 0.5 | 0.9                   | 0.3 | 1.2 | 1.5                   | 1.4 | 0.3 | 0.7 | - | 5  | 34 | -               | 2.5 | 3.3            | 1.4 | 1.2 | 1.2 |
|      | Average                    | 0.7 | 0.7 | 0.7                   | 1.3 | 1.2 | 0.9                 | 1.1 | 8.0 | 1.0                   | 0.6 | 0.5 | 1.2                   | 1.1 | 0.7 | 8.0 | - | 27 | 43 | -               | 1.5 | 2.7            | 1.1 | 1.1 | 1.1 |

Figure D1. Final calibrated parameter values for all catchments across each model configuration (BL, GW, GW-RC).

We appreciate your constructive feedback and the time you have dedicated to reviewing our work. Sincerely,

Frédéric Talbot on behalf of all authors

185